# AircraftVerse: A Large-Scale Multimodal Dataset of Aerial Vehicle Designs

**Adam D. Cobb,**[*]    **Anirban Roy,**[*]    **Daniel Elenius,**[*]    **F. Michael Heim,**[†]    **Brian Swenson,**[‖]

**Sydney Whittington,**[‖]    **James D. Walker,**[†]    **Theodore Bapty,**[‡]    **Joseph Hite,**[‡]    **Karthik Ramani,**[§]

**Christopher McComb**[¶]    **Susmit Jha**[*]

[*] Computer Science Laboratory, SRI International
[†] Mechanical Engineering Division, Southwest Research Institute
[‖] Intelligent Systems Division, Southwest Research Institute
[‡] Institute of Software Integrated Systems, Vanderbilt University
[§] School of Mechanical Engineering, Purdue University
[¶] Department of Mechanical Engineering, Carnegie Mellon University

## Abstract

We present `AircraftVerse`, a publicly available aerial vehicle design dataset. Aircraft design encompasses different physics domains and, hence, multiple modalities of representation. The evaluation of these cyber-physical system (CPS) designs requires the use of scientific analytical and simulation models ranging from computer-aided design tools for structural and manufacturing analysis, computational fluid dynamics tools for drag and lift computation, battery models for energy estimation, and simulation models for flight control and dynamics. `AircraftVerse` contains 27,714 diverse air vehicle designs - the largest corpus of engineering designs with this level of complexity. Each design comprises the following artifacts: a symbolic design tree describing topology, propulsion subsystem, battery subsystem, and other design details; a STandard for the Exchange of Product (STEP) model data; a 3D CAD design using a stereolithography (STL) file format; a 3D point cloud for the shape of the design; and evaluation results from high fidelity state-of-the-art physics models that characterize performance metrics such as maximum flight distance and hover-time. We also present baseline surrogate models that use different modalities of design representation to predict design performance metrics, which we provide as part of our dataset release. Finally, we discuss the potential impact of this dataset on the use of learning in aircraft design and, more generally, in CPS. `AircraftVerse` is accompanied by a data card, and it is released under Creative Commons Attribution-ShareAlike (CC BY-SA) license. The dataset is hosted at https://zenodo.org/record/6525446, baseline models and code at https://github.com/SRI-CSL/AircraftVerse, and the dataset description at https://aircraftverse.onrender.com/.

## 1   Introduction

Datasets of complex cyber-physical systems (CPS) are difficult to build and large CPS datasets are not publicly available. Their availability is limited for multiple reasons, such as: proprietary restrictions; difficulty in assembling the right mix of experts; and the slow manual design process

37th Conference on Neural Information Processing Systems (NeurIPS 2023) Track on Datasets and Benchmarks.

due to their complexity. However, a huge opportunity exists to apply data-driven approaches to CPS once such a dataset becomes widely available. Electric Vertical Take-Off and Landing (eVTOL) aircraft represent an emerging class of CPS. The use of electrical propulsion and the growing energy density of available batteries have fueled rapid growth in eVTOL aircraft designs from food delivery/logistics [12], to the detection of sharks along highly populated beaches [18], and to air taxis. We expect this variation to only increase as battery technology continues to improve [10]. The diversity within the electric aerial vehicle design space is large due to the many choices available for selecting structural, mechanical, and electrical components. The multiphysics nature of CPS designs necessitate heterogeneity in representation, such as the use of stereolithographic (STL) files for computational fluid dynamics (CFD) analysis and the use of symbolic descriptions of motors and speed controllers for electrical analysis. Data-driven learning methods for CPS must be able to handle this diversity and multimodality of CPS designs. `AircraftVerse` provides such a dataset to enable further research into data-driven methods for characterizing or designing CPS.

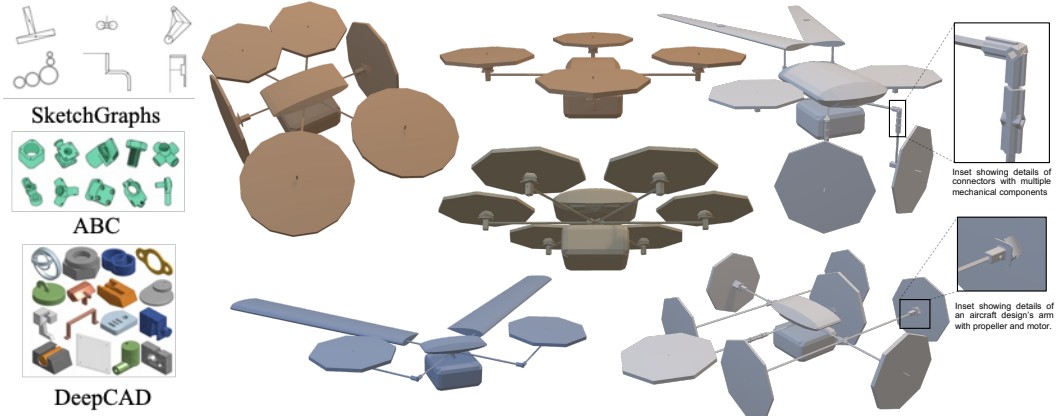

(a) CAD Datasets        (b) Example of 3D CAD models of aircraft designs in `AircraftVerse`

Figure 1: Existing CAD datasets (SketchGraphs [28], DeepCAD [35], ABC [19]) are focused on CAD for mechanical parts. Aircraft designs in `AircraftVerse` include CAD models as one of the modalities. A CAD model (STEP or STL) is an assembly of several components such as propellers, wings, connectors, beams, motors, batteries, and hubs. The CAD designs in `AircraftVerse` are more complex compared to the existing datasets. The inset magnifications of a couple of parts of one of the designs in the figure above demonstrates this complexity. In addition to CAD models, each design also includes a symbolic design tree with additional details such as propulsion and battery subsystems that are needed for performance analysis (e.g. electrical and flight dynamics analysis). `AircraftVerse` also contains the result from the evaluation of each design using high-fidelity scientific and engineering tools. Thus, `AircraftVerse` is a CPS dataset and not just a CAD dataset.

`AircraftVerse` (Figure 1) contains **27,714 diverse aircraft designs** where each design is represented in multiple modalities, including a computer aided design (CAD) model and symbolic description. Each design is also accompanied with detailed evaluation results. Thus, `AircraftVerse` is naturally amenable to emerging neurosymbolic and other deep learning methods for generative modeling, surrogate learning and sequence-to-sequence models. Overall, our paper is structured as follows. In Section 2 we provide a summary of the few existing related datasets. We then introduce `AircraftVerse` in Section 3 and describe the constituent parts that make up a design. We highlight the potential of `AircraftVerse` in Section 4 by providing an overview of the diversity of designs and by displaying experiments on surrogate modeling, while noting that this covers just a small part of what kind of experimentation is possible with this new dataset. We then conclude in Section 7.

## 2 Related work

The use of machine learning in CAD has gained significant attention, and a few datasets have been proposed in recent literature to enable development and benchmarking of machine learning approaches. SketchGraphs dataset [28] is a collection of sketches extracted from parametric CAD models which begin as two-dimensional (2D) sketches consisting of geometric primitives (e.g., line

segments, arcs) and explicit constraints between them (e.g., coincidence, perpendicularity) that form the basis for three-dimensional (3D) construction operations. This dataset has been used for generative model of CAD sketches [34], and other applications of learning in physical design [29, 23]. Another example of a CAD dataset that is focused on physical structure is SimJEB [33], which is a dataset of crowdsourced mechanical brackets and accompanying structural simulations. DeepCAD [35] is a dataset of 3D shapes corresponding to objects such as flanges, pipes and screws, represented as a sequence of operations used in a CAD framework to generate these shapes. Another dataset for 3D engineering shapes is the ABC dataset [19], which comprises geometric models, each defined by parametric surfaces and associated with ground truth information on the decomposition into patches. These datasets are excellent resources for their target application domains such as extrapolating 2D sketch to CAD designs, and generating mechanical parts.

In contrast, `AircraftVerse` is a dataset that covers the more complex design space of electric aircraft focused on system-level CPS design and thus, complements existing CAD datasets. In Figure 1, we illustrate the complexity of the aircraft designs in `AircraftVerse` compared to mechanical components from the existing CAD datasets. The CAD models of an aircraft design in `AircraftVerse` are an assembly of several mechanical components, such as propellers, motors, wings, hub and connectors. Further, the `AircraftVerse` dataset includes additional description of a design beyond its CAD model, such as its propulsion subsystem and its electrical subsystem. This is crucial for creating a description that can predict the performance (flight dynamics, electrical analysis) of a design as designs with similar CAD structure can have very different performances depending on the used components. We also provide the performance of each design using detailed high-fidelity physics and engineering simulation tools to enable the use of this dataset not just for generative modeling, but also for learning surrogates and design characterization and optimization.

## 3 `AircraftVerse` **dataset**

The key characteristics of `AircraftVerse` are as follows:

- The number of designs in `AircraftVerse` is 27,714 making it a uniquely **large-scale CPS dataset** with high design complexity. Design curation required finding valid aircraft configurations followed by detailed simulations to evaluate its performance objectives. We have attempted to create a balanced dataset that includes a sufficient number of aircraft designs across a range of different flight performance metrics, such as hover times and maximum flight distances. Our search for designs included over a hundred thousand candidates from which we selected these 27,714 designs to ensure diversity in design and performance. Our design process itself is diversity-preserving and ensures optimization does not lead to very similar designs [5, 6].

- `AircraftVerse` represents design using **multiple modalities** that include: a symbolic design tree describing the design topology, propulsion subsystem, battery subsystem, and other design details; a STandard for the Exchange of Product (STEP) model that is a decomposable CAD model showing each part separately; and a stereolithographic (STL) CAD model that is ideal for computational fluid dynamics analysis, and the corresponding 3D point cloud for the shape of the design.

- `AircraftVerse` uses **results from high-fidelity physics models** and an evaluation pipeline developed as a part of DARPA's Symbiotic Design of CyberPhysical System's program[1] to evaluate different aircraft designs. These physics models characterize performance metrics such as maximum flight distance and hover-time. The evaluation pipeline uses a mixture of custom flight dynamics simulators [31] and commercial tools such as Creo [24]. The evaluation of a design to determine performances such as its drag, lift, maximum flight time and hover time requires a significant compute infrastructure and requires subject-matter expertise. We include these evaluation results as part of the dataset.

- The designs in `AircraftVerse` exhibit a very **high degree of diversity** in their topology, the choice of energy subsystem and the choice of propulsion. The designs are a mixture of rotorcrafts, winged aircrafts and hybrids with 28% capable of vertical takeoff and landing.[2] To the best of our knowledge, there is no other available corpus of such diverse aircraft designs with the design details and the results from detailed scientific and engineering evaluation.

---

[1]https://www.darpa.mil/program/symbiotic-design-for-cyber-physical-systems

[2]We note that some aerial vehicles, such as many fixed wing aircraft, are launched (or catapulted) and therefore are not required to hover.

- We include multiple **baseline surrogate models** to predict some of the key design performance metrics. Our surrogate models include a transformer encoder model and an LSTM model that take the symbolic design configuration, as its input, as well as a graph convolution neural network and a PointNet model that use the point cloud modality. These baseline models are used to predict the mass, the maximum flight distance, the maximum hover-time and the presence of any structural interferences that needs to be avoided for fabrication and manufacturing.

**Dataset Availability.** The dataset is publicly available for free under Creative Commons Attribution-ShareAlike (CC BY-SA) license, which will enable future extension and adaptation of the dataset by others. With respect to its maintainability and long-term availability, the dataset is hosted at `https://zenodo.org/record/6525446` and will be maintained by SRI International[3], a non-profit research institute with over 75 years of history of contributing to society and research community, with a proven track record of building, maintaining and distributing several datasets such as BioCyc [16], Voices [27], and open-source tools[4] - some of which have been maintained for multiple decades.

**Dataset Curation.** The designs in `AircraftVerse` are battery-powered, where the propulsion comes from the electric motors that power the propellers. Our design corpus includes a range of propellers, motors, batteries, and fixed wings. The list of components is provided in Appendix F. We use the propulsion subsystem design as an example to describe the curation of the dataset. We use a combination of wings and propellers (in the right topology) provides the thrust and lift needed for efficient horizontal flight and vertical take-off and landing. Our components are diverse, such as the propellers have a different number of rotor blades with different diameters and pitches, the component motors have different $K_v$, $K_m$, $K_t$ ratings, and the batteries have different peak and continuous current ratings and different voltage ratings. Consequently, the design choices allow significant diversity in structure and composition for similar performance. For example, the thrust produced by a propeller-motor pair can be related approximately to the parameters by the following formula: $Thrust \propto (K_v \times Voltage)/(Pitch \times Diameter)$, where $K_v$ is the motor's "Kilovolt" rating, which represents the number of revolutions per minute (RPM) that the motor will produce per volt, $Voltage$ is the voltage being applied to the motor, $Pitch$ is the distance that the propeller would move forward in one rotation, and $Diameter$ is the distance across the propeller. Our design used more detailed models [31, 24], but this dependency illustrates how the same thrust can be produced by different combinations of propellers and motors. High $K_v$ rating motors that rotate much faster can produce the same thrust with smaller propellers when compared to lower $K_v$ (slower) motors with larger diameter propellers. These combinations can have different continuous and peak current requirements that would influence the battery choice. In addition to these propulsion choices, connectors such as parametric joints, hubs, and arms can be put together in flexible ways to build interesting design geometry and topologies. These can have an impact on the experienced drag, which would then influence the requirements for propulsion. Thus, the design of aircraft requires making a number of tightly coupled design choices. The performance of each design can be assessed using a pipeline comprising commercial tools such as Creo [24] and custom flight dynamics model (FDM) [31, 2]. Each aircraft is also assessed on controllability (existence of trim states - where the aircraft remains stable in the absence of environment perturbations) at different speeds using the FDM. In particular, the FDM evaluates whether the aerial vehicle can fly at a specific velocity through optimization schemes that check if the translational and rotational accelerations can be driven to zero for a given design during vertical take-off and horizontal flight (more details in Appendix C). Manually finding a large number of feasible designs is very challenging and prohibitively time-consuming. Instead, we use a learning-based design approach [6]. We create a number of designs using a procedural aircraft generator that uses heuristics provided by domain experts, such as limiting the complexity of produced designs to have at most 16 propellers and at most 12 wings, using design motifs and symmetries that help with controllability of the aircraft. A transformer-based model is trained on designs with good flight characteristics to act as a filter in the future generation process (Appendix E, [6]). Each of the final 27,714 designs included in the dataset is run through the detailed scientific and engineering models [31, 2, 24] to generate the metadata describing the design performance and characteristics.

**Design Structure and Representation.** Each design in the dataset consists of the following:[5]

---

[3]https://en.wikipedia.org/wiki/SRI_International

[4]https://github.com/SRI-CSL

[5]For a complete printout see Appendix G.

- `design_tree.json`: The design tree describes the design topology, choice of propulsion and energy subsystems. The tree also contains continuous parameters such as wing span, wing chord, and the lengths of propeller arms and connectors. In our dataset, we also include a preorder traversal of the design tree and store this as `design_seq.json` to facilitate the use of sequence-friendly model architectures. In addition, we include `design_low_level.json`, which is a more fine-grained engineering representation of the design. This fine-grained representation includes significant repetition that is avoided in the abstract tree representation through the use of symmetry (such as specifying that the same arm structure is repeated six times around a hub). See the pictorial representation of the design tree in Figure 2 that demonstrates a simple example of such a symmetry.

- `Geom.stp`, `cadfile.stl`, and `pointCloud.npy`: CAD design for the aerial vehicle in compositional STEP format (ISO 10303 standard), its stereolithographic STL file, and a generated pointcloud of 10,000 points from the CAD representation.

- `output.json` and `trims.npy`: Summary files containing the vehicle's performance metrics such as maximum flight distance, maximum hover time, flight distance at maximum speed, maximum current draw, and mass. The `trims.npy` contains the [*Distance, Flight Time, Pitch, Control Input, Thrust, Lift, Drag, Current, Power*] at each evaluated trim state (velocity) of the aircraft. Designs that have more trim states and with contiguous trim states are preferable as these designs would be relatively easier to control even when the environment is noisy. Thus, both the `output.json` and `trims.npy` contain the evaluation of the performance of each design. See Figure 2 for an example of these evaluation indicators.

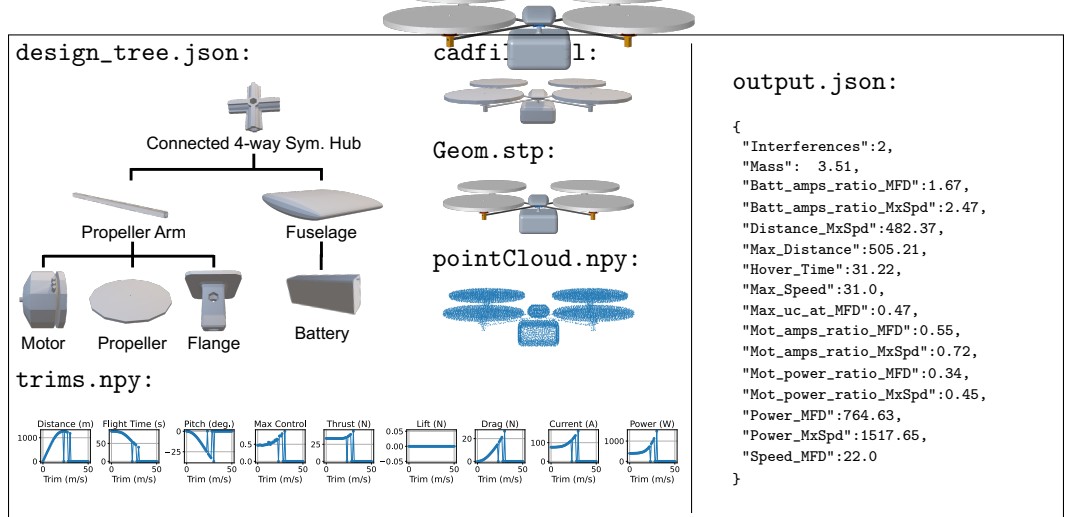

Figure 2: An illustration of all the key file components that make up an aircraft design. The `design_tree.json` captures all aspects of the design (e.g. 4-way hub) that appears in both the `cadfile.stl` and the `Geom.stp`. Additional files of `design_seq.json` and `design_low_level.json` are directly derived from the design tree. The performance of the aerial vehicle is displayed in both the performance file labeled `output.json` and in the `trims.npy`.

**Design Example.** The design tree used for aircraft designs reflect the hierarchical nature of CPS designs. Figure 2 shows an example design. It has a *4-way central hub* component. Each of the four arms of the hub can have a propulsion subsystem attached to it. We identify this subsystem as a *MainSegment* which can be implemented via different choices (subtrees expanding *MainSegment*) such as *BendSegment*, *PropArm*, *ExtendedPropArm*, *WingArm*, and so on. Each of these choices describe the nature of the propulsion subsystem, such as whether we are using propeller or wing, and whether the arm making connection is a simple connector or a composition of connectors. Further, some of these subsystems can be recursive, allowing fractal growth of the design. For example, an arm itself might contain another segment with a hub having a number of arms. Our symbolic representation of design also enables exploitation of symmetry to compress the design description, and inclusion of expert knowledge in the design. For example, if we want all four arms of a hub to contain the same subsystem, then we only need to specify it once (as a single child in the design tree) and use a symmetry tag over the hub, for example, *ConnectedHub4_Sym* denotes a hub with four connections - each having the same subsystem. In the case of Figure 2, a propulsion subsystem

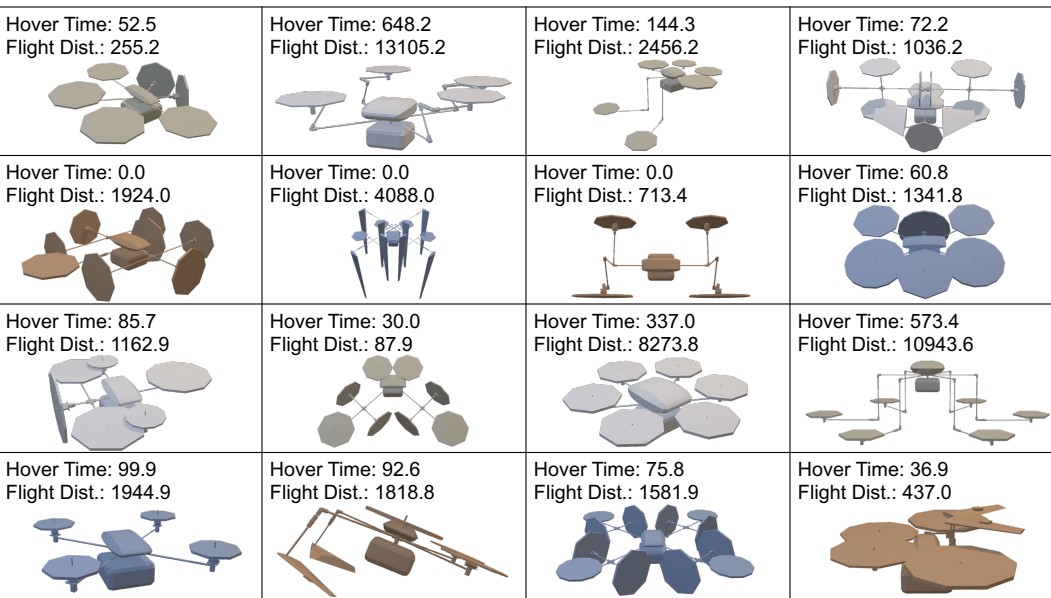

| Hover Time: 52.5
Flight Dist.: 255.2 | Hover Time: 648.2
Flight Dist.: 13105.2 | Hover Time: 144.3
Flight Dist.: 2456.2 | Hover Time: 72.2
Flight Dist.: 1036.2 |
|---|---|---|---|
| Hover Time: 0.0
Flight Dist.: 1924.0 | Hover Time: 0.0
Flight Dist.: 4088.0 | Hover Time: 0.0
Flight Dist.: 713.4 | Hover Time: 60.8
Flight Dist.: 1341.8 |
| Hover Time: 85.7
Flight Dist.: 1162.9 | Hover Time: 30.0
Flight Dist.: 87.9 | Hover Time: 337.0
Flight Dist.: 8273.8 | Hover Time: 573.4
Flight Dist.: 10943.6 |
| Hover Time: 99.9
Flight Dist.: 1944.9 | Hover Time: 92.6
Flight Dist.: 1818.8 | Hover Time: 75.8
Flight Dist.: 1581.9 | Hover Time: 36.9
Flight Dist.: 437.0 |

Figure 3: Diverse vehicle designs in `AircraftVerse` with different structures and with different performances. Some designs can hover in place. Hover time is in seconds and flight distance is in meters.

comprises a propeller, motor, and a flange. Further, the hub also connects to a fuselage that contains a battery subsystem that can consist of a single or dual battery subsystem along with additional electronics and their position within the fuselage. An explicit example of this is provided in Appendix G. Additionally, the full corpus of options as included in Appendix F is also provided as a Python dictionary as part of `AircraftVerse`.

**Design evaluation.** For each design, we use state-of-the-art tools [24, 2] to compute the physical characteristics, such as the mass and the component interferences, as well as to render the 3D CAD models (STEP and STL files that are included in `AircraftVerse`). These details are then passed to the FDM [31] for evaluating flight dynamics (more details in Appendix C). The dynamics model considers the aircraft velocity and its roll, pitch and yaw rotations, along with the electrical state such as the power drawn from the battery and motor current, which in turn determine the torque on the propeller and its generated thrust. The dynamics model summarizes the impact of different forces such as gravity, drag, propeller lift and wing lift, and includes scientific models for conversion of electrical energy into mechanical energy in the propulsion subsystem. From these evaluations, we include `Batt_amps_ratio_MFD`, `Batt_amps_ratio_MxSpd` that correspond to the ratio of maximum current drawn by the design compared to the maximum current rating of the battery at maximum flight distance and maximum speed, respectively. These indicate the robustness of the battery subsystem of the design. We also include `Mot_amps_ratio_MFD`, `Mot_amps_ratio_MxSpd`, `Mot_power_ratio_MFD`, `Mot_power_ratio_MxSpd` that represent the ratio of the current/power in the motor divided by the maximum allowed current/power corresponding to the maximum flight distance or maximum speed of the vehicle. These indicate the robustness of the propulsion subsystem of the design, its endurance, and its agility. Finally, we include design metrics such as the flight distance at maximum speed (`Distance_MxSpd`), maximum flight distance (`Max_Distance`), and the maximum hover (`Hover_Time`).

## 4 Experiments: diversity, multimodality and baseline models

**Diversity.** `AircraftVerse` includes a diverse set of aircraft designs (samples shown in Figure 3), and the multimodal representation makes this dataset relevant to a variety of challenge problems in machine learning. The components of a design in `AircraftVerse` are selected from a large component corpus with hundreds of choices of propellers and motors (Appendix F). In addition, each discrete choice of component comes with its own set of attributes that fall in their own ranges (e.g.

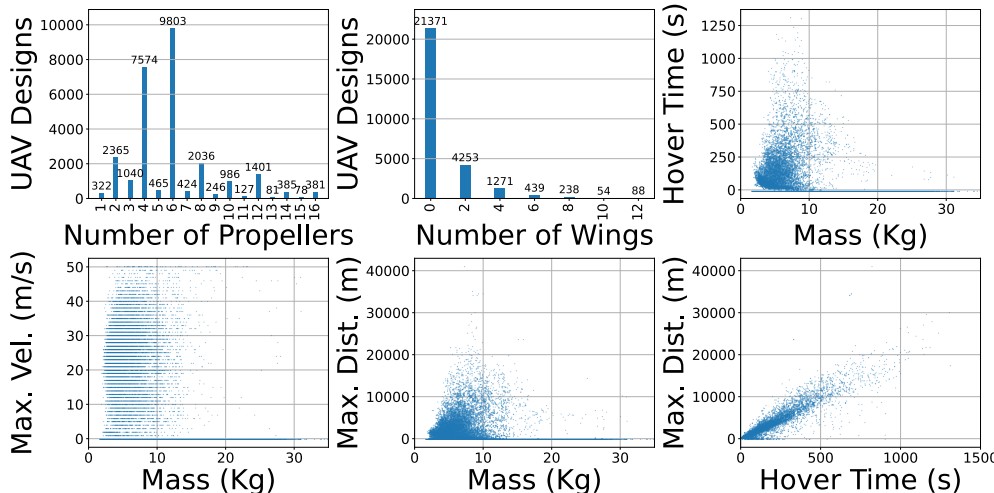

Figure 4: The designs in `AircraftVerse` exhibit diverse performance with respect to characteristics such as maximum flight distance and hover-time. They also have very different physical characteristics such as mass, number of propellers, and number of wings.

motors have a large set of parameters such as motor armature-winding resistance and inductance, mass, counter-electromotive force constant, and motor torque constant). Adding to the overall complexity, there are also the continuous parameters. For example, structural components such as connectors have their radius and length as continuous parameters. In Appendix F, we identify the attributes of the key components in an aerial vehicle design and whether their values are fixed or variable within a range. Some characteristics of the design diversity are captured in Figure 4. The figure highlights some of our curation choices. First, the majority of designs have an even number of propellers due to the symmetry built into our tree structure definition. Second, our dataset precludes designs without propellers and favors designs with fewer wings. Finally, while a significant number of designs can take off and achieve horizontal flight (with a large variation in ability), one of the challenges in building a diverse dataset comes at the expense of many designs failing to have the hovering capability or other eVTOL flight performance metrics. However, these designs are useful for learning potential rules as to why these designs have poor characteristics compared to others, so we also include these in the dataset.

While structural differences lead to visually striking diversity, we also highlight that design diversity has other aspects, such as the use of different components within the same design topology. Vehicle designs with the same topology can perform very differently. Figure 5 contains four symmetric quadcopters with the same topology (same design tree structure). However, their performance metrics differ significantly, with the maximum flight distance differing by a factor of over 2x. Additionally, the electronics of the rightmost quadcopter varies from the other three in that it has two batteries rather than one. Thus, the performance of designs with similar topologies cannot be predicted from just from the CAD design, but requires the symbolic design description in `design_tree.json`.

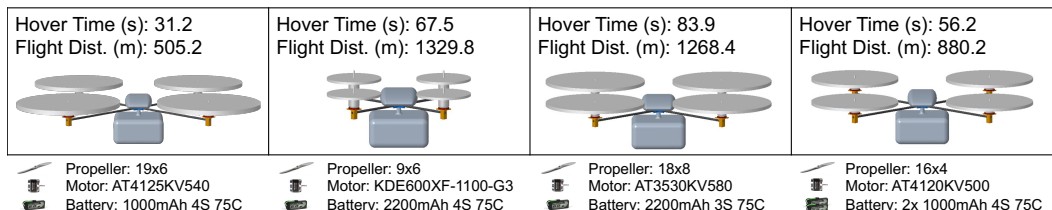

Figure 5: Diverse aerial vehicle designs in `AircraftVerse` with same topology structure but different motor/propeller/battery choices leading to diverse design performance.

**Multimodality.** The `AircraftVerse` uses multiple modalities to represent an aircraft design, as illustrated earlier in Figure 2. We would not expect the performance of electric aerial vehicle designs to be

accurately predicted on the basis of only one of the modalities (e.g. the 3D CAD model). Recent work on vision-language models [8, 15, 1, 20, 7] would be especially relevant to large CPS datasets such as `AircraftVerse`. However unlike in the vision-language scenario, `AircraftVerse` has stricter metadata that enforces physical and electrical constraints. The multi-modality of `AircraftVerse` makes it applicable to a wide class of machine learning models. `AircraftVerse` includes the following modalities:

- Structured sequential data: The success of transformer-based sequence models used in natural language [30, 9, 26, 3, 21] has driven their adoption and extension to structured data such as computer programs. The sequential data from the design trees in `AircraftVerse` is an interesting use case for transformer models. Further, the designs in `AircraftVerse` are more structured than natural language text as the design components need to satisfy physical and electrical constraints[6].

- 3D CAD models: There is a growing interest in using machine learning for CAD design [28, 34, 23, 33, 35, 19]. The 3D CAD models (STL and STEP files) in `AircraftVerse` provide a large dataset with complex 3D shapes. The metadata associated with the designs can be used to train and evaluate methods that predict the physics metrics from the CAD models.

- 3D Pointcloud: The use of machine learning for learning generative models or manipulating or segmenting 3D point clouds has also received significant attention. `AircraftVerse` includes 3D point clouds extracted from the STL and STEP files. Thus, this dataset with the pointclouds can be used as a benchmark for 3D point cloud classification (e.g., presence or absence of interference) [36, 4, 13] and regression (e.g., predicting flight characteristics).

**Baseline Models.** We describe baseline models applied multiple modalities of `AircraftVerse`.[7] We use a transformer encoder (T. Enc.) [30] model and a LSTM [14] model to predict a set of design performance characteristics of an aircraft from its symbolic design tree description. In our example, even using these symbolic descriptions of a design contain information from multiple modalities, as these sequences include electrical, mechanical, and topological information. We also test out the 3D modality of the design representation by applying both a graph convolutional neural network (GCNN) [17, 32] and PointNet [25] to the point clouds. For the sequences, we use the design representation in `design_seq.json` and use our custom-built tokenizer to convert each token of a design sequence from the symbolic representation (e.g. {'node_type': 'ConnectedHub3_2_1'}) into a tensor. For our embedding, we assign one-hot encoding for the keys and an additional one-hot encoding for the values. For keys that require floats as values (e.g. 'armLength') we ensure that there is a class that corresponds to the float in the value embedding, as well as appending the value of the float to the tensor. Each token of the sequence is a 749-element vector corresponding to the concatenation of 43 classes of keys, 673 classes of values, 32 possible attributes of electrical/mechanical components, and a final dimension for float values. We provide our tokenization code as part of the dataset release, as well as the baseline models. Further architectural details of the models are presented in Appendix D.

**Results.** The results of predicting the performance statistics from the design sequences are presented in Table 1. This table shows how the sequential design description, containing specific component information such as mass and electrical information, is a useful modality for certain metrics such as the ability of a design to fly. However, both the baseline GCNN model and PointNet are better at inferring structural interferences between components as it directly incorporates 3D information. Future approaches that combine the 3D structural information and symbolic design descriptions may be necessary and we have demonstrated here that the `AircraftVerse` dataset includes all of these informative modalities. Figure 6 shows both the receiver operator characteristic (ROC) curve and precision-recall curve for the three models. Again, highlighting the superior performance of the sequential information in estimating the high level CPS performance metrics such as ability to hover.

## 5   Limitations of `AircraftVerse`

While `AircraftVerse` presents itself as a unique CPS dataset with a high level of complexity, there are several limitations to using this dataset. One limitation is that aircraft designs are constrained to fit within the tree-structured representation. This tree structure currently means all designs start from a sampled single hub and are then expanded. This process is described in `https://github`.

---

[6]See Appendix F for a definition of the structure of our design trees and sequences.

[7]Our models were trained using an NVIDIA A100 SXM4 80GB GPU.

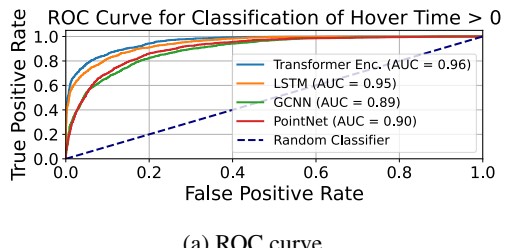
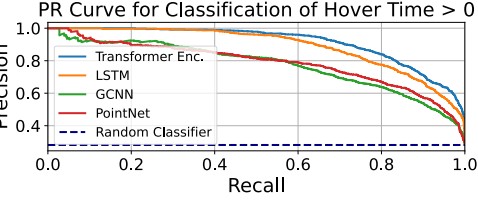

(a) ROC curve          (b) Precision-Recall

Figure 6: ROC and precision-recall curves for predicting a design's ability to hover. These results suggest that the symbolic design description in `AircraftVerse` can be used to predict flight characteristics, whereas Table 1 highlights the utility of point clouds for predicting structural information.

Table 1: Specification prediction using baseline models.

| Approach | Data | Hover (Acc. ↑) | Hover Time (R$^2$ ↑) | Mass (R$^2$ ↑) | Flight Distance (R$^2$ ↑) | Interference (F1 Score ↑) |
|---|---|---|---|---|---|---|
| T. Enc. | Sequence | **0.9004** | **0.6778** | 0.9964 | **0.6900** | 0.8270 |
| LSTM | Sequence | 0.8838 | 0.6189 | **0.9974** | 0.3467 | 0.8184 |
| GCNN | Point Cloud | 0.8380 | 0.3936 | 0.8888 | 0.4486 | 0.9758 |
| PointNet | Point Cloud | 0.8474 | 0.4571 | 0.9212 | 0.5099 | **0.9848** |

`com/SRI-CSL/AircraftVerse/tree/main/prob_gen`, which also includes all incorporated prior domain knowledge. Designs are also limited by the available components in the database. For example, hybrid (gas and electric) vehicles are not possible. However, all of the database components are from real-world catalogues which increases the overall value of the dataset.

As with all scientific simulators, the flight dynamics model is limited in its ability to represent all aspects of the real-world. As such, there are likely corner cases that would cause unrealistic behaviors. One weakness of the simulator is in the drag model as described in Equation (4) (Appendix C). The drag model approximates the total surface area in each axis and therefore would be superseded by higher fidelity CFD software. However, the simulator uses conservative estimates for such calculations and has been compared to higher fidelity models to ensure they capture the correct behavior. The results are therefore still useful for design prototyping purposes. A further extension to the simulator to increase fidelity would also be to model material properties such as stress and strain during flight. Further development will continue to address these simulation gaps.

Given these limitations we still highlight the value of this dataset as useful for machine learning researchers looking to explore a novel CPS dataset as well as domain experts looking to explore the potential of data-driven approaches for design. This is precisely what we found to be the case throughout the collaboration that led to this work. As a result, we believe that `AircraftVerse` is sufficiently complex and interesting for both sets of researchers, who would see value in the data, while ensuring that they take into account all the limitations.

## 6 Ethics Statement

The data included in the `AircraftVerse` dataset does not include any personal or sensitive information. It is a technical dataset of air vehicle designs, and thus it does not raise any direct ethical issues regarding privacy or data protection. The dataset is intended for academic and research use and is released under a Creative Commons Attribution-ShareAlike (CC BY-SA) license. This license allows for sharing, copying, distributing, and transmitting the work, as well as adapting the work and making commercial use of the work under the condition that appropriate credit is given, a link to the license is provided, and any changes made are indicated. This licensing framework is intended to promote open access to information and collaboration in the scientific community, while still recognizing and respecting the intellectual property rights of the creators.

Given the potential wide-ranging impact of this dataset on the use of learning in aircraft design and other CPS, it is crucial that future research using this dataset is conducted with careful consideration

of potential ethical implications. This includes not only the avoidance of misuse but also consideration of environmental impacts, safety, and other societal implications. We welcome community input on the dataset, its uses, and its potential ethical implications. We want to promote responsible use and want to address potential ethical concerns proactively.

## 7 Conclusion

We have introduced a new multi-modal CPS dataset that will enable researchers to explore a new interdisciplinary area of application for machine learning. We emphasize the richness of `AircraftVerse`, where we provide 27,714 aircraft designs, each with associated metadata summarizing the performance of the design. We highlight that our experiments and baseline models have only touched the surface of the different approaches and problems that can be explored with this dataset due to the availability of all the metadata associated with each design. In addition to the material presented here, we also include an extensive supplementary materials section as well as a website: `https://aircraftverse.onrender.com/`, a GitHub repo: `https://github.com/SRI-CSL/AircraftVerse/`, and the dataset release at `https://zenodo.org/record/6525446`.

## Acknowledgments and Disclosure of Funding

This material is based upon work supported by the United States Air Force and DARPA under Contract No. FA8750-20-C-0002. Any opinions, findings and conclusions or recommendations expressed in this material are those of the author(s) and do not necessarily reflect the views of the United States Air Force and DARPA.

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

# A  Data Card

In this section we include the data card that follows the structure in Pang and Lee [22] as highlighted in Gebru et al. [11].

## A.1  Motivation

**For what purpose was the dataset created?**  The dataset was created to enable research on designing cyber physical systems. In particular, we wanted to see if it was possible to predict characteristics of aircraft designs, and be able to filter out poor performing designs before needing to use expensive scientific models. In order to achieve this, we needed to build a large representative dataset to train our models.

**Who created the dataset?**  The dataset was created by the authors: Adam D. Cobb, Anirban Roy, Daniel Elenius, and Susmit Jha, at SRI International; Brian Swenson, Sydney Whittington, and James Walker at Southwest Research Institute; Theodore Bapty, and Joseph Hite at Vanderbilt University; Karthik Ramani at Purdue University; Christopher McComb at Carnegie Mellon University; supported by DARPA under the Symbiotic Design for Cyber-Physical Systems program with contract FA8750-20-C-0002.

**Who funded the creation of the dataset?**  DARPA under the Symbiotic Design for Cyber-Physical Systems program with contract FA8750-20-C-0002.

## A.2  Composition

**What do the instances that comprise the dataset represent?**  The instances are aerial vehicle designs as generated by us and evaluated on scientific models [2, 31]. We also provide metadata in the form of the corpus dictionary such that the attributes of the propellers, motors, batteries, and wings are available.

**How many instances are there in total**  There are 27,714 instances in total.

**Does the dataset contain all possible instances or is it a sample of instances from a larger set?**
The dataset is a sample of instances that were generated according to our grammar. As our generator includes both discrete and continuous parameters, it is not possible to capture all possible instances of the designs. Furthermore, designs are allowed to contain recursively generated components, such as arms, such that all possible topologies cannot be sampled. The plots in Figure 4 show the diversity in designs.

**What does each instance consist of?**  Each instance consists of a design tree, an interpreted design sequence, a low level design sequence, an STL file (3D model), a npy file containing an array of trims, a npy file containing a point cloud, and a JSON file reporting flight statistics. Please refer to 2

**Is there a label or target associated with each instance?**  The labels are given via the output.json and trims.npy files. These files contain labels such as whether a design has interferences and if they are able to fly. It is also possible to use the design tree or design sequence to label the 3D data. For example the low level design sequence can be used to label how many propellers are in the aircraft design or to predict which battery has been selected based on the performance file.

**Is any information missing from individual instances?**  Everything is included. No data is missing.

**Are relationships between individual instances made explicit?**  Individual instances are sampled independently from our generator and are therefore not explicitly related. The corpus dictionary can also be used to determine the attributes of the available components where appropriate, where this corpus is relevant to all designs.

**Are there recommended data splits (e.g., training, development/validation, testing)?** In the notebook code we provide our training, validation, and test splits but they are not explicitly recommended as the data has multiple use-cases.

**Are there any errors, sources of noise, or redundancies in the dataset?** The only errors come from the available fidelity of the scientific models. Please refer to [2, 31] for the limitations of the models. As a result, some of the outputs from the metadata contain NaNs. These are simple enough to filter out from the dataset where needed.

**Is the dataset self-contained, or does it link to or otherwise rely on external resources?** The dataset is entirely self-contained.

**Does the dataset contain data that might be considered confidential?** No.

**Does the dataset contain data that, if viewed directly, might be offensive, insulting, threatening, or might otherwise cause anxiety?** Not that the authors are aware of.

**Does the dataset identify any subpopulations (e.g., by age, gender)?** Not applicable.

**Is it possible to identify individuals (i.e., one or more natural persons), either directly or indirectly (i.e., in combination with other data) from the dataset?** Not applicable.

**Does the dataset contain data that might be considered sensitive in any way?** No.

## A.3 Collection Process

**How was the data associated with each instance acquired?** The data was generated by sampling our probabilistic design generator and then compiling each design into a format that was suitable for the scientific models. The scientific models provided performance metrics which we then curated into the `output.json` files and the STL files.

**What mechanisms or procedures were used to collect the data?** Each design took approximately 3 minutes to evaluate. As a result we estimate that it took almost 60 days of compute power to build `AircraftVerse`. As part of the pipeline, we required access to a CREO license [24].

**If the dataset is a sample from a larger set, what was the sampling strategy?** The dataset was sampled from our probabilistic design generator and was therefore a stochastic process.

**Over what timeframe was the data collected?** The data was generated using four compute nodes and therefore took over 2 weeks to generate. However, this was part of a process that took around two years of development to get to. Part of that process was building domain knowledge across the collaboration and building up the right compute architecture.

**Were any ethical review processes conducted?** Not applicable.

**Did you collect the data from the individuals in question directly, or obtain it via third parties or other sources (e.g., websites)?** Not applicable.

**Were the individuals in question notified about the data collection?** Not applicable.

**Did the individuals in question consent to the collection and use of their data?** Not applicable.

**If consent was obtained, were the consenting individuals provided with a mechanism to revoke their consent in the future or for certain uses?** Not applicable.

**Has an analysis of the potential impact of the dataset and its use on data subjects been conducted?** Not applicable.

## A.4 Preprocessing/cleaning/labeling

**Was any preprocessing/cleaning/labeling of the data done?**  For the machine learning models, we processed the design trees into a unique key-value sequence as described in Section 4. As part of evaluating the performance of each design, we recorded the performance metrics in the `output.json` file and the `trims.npy` file. We also used a combination of data-driven heuristic approaches and hard rules to remove certain designs from the final dataset. We limited designs to a maximum of 12 wings and 16 propellers. We further added a data-driven preprocessing step to remove designs that would definitely not hover. We were careful to select a threshold that did not sacrifice diversity.

**Was the "raw" data saved in addition to the preprocessed/cleaned/labeled data**  Each design contains the raw data as well as the processed sequences. We consider the design trees, the `STL` files and the performance files to be raw data.

**Is the software that was used to preprocess/clean/label the data available?**  The software required to label the data requires a CREO license along with proprietary software owned by Southwest Research Institute. It is possible that the latter software will be released in the future.

## A.5 Uses

**Has the dataset been used for any tasks already?**  At the time of publication, the only use-case of our dataset can be seen in Section 4.

**Is there a repository that links to any or all papers or systems that use the dataset?**  Not applicable at this time as this is the first instance of the data release.

**What (other) tasks could the dataset be used for?**  The dataset could be used for anything related to modeling or understanding the behavior of aircraft. For example one could envision learning rules for good designs, inverting the design by predicting sequences from point clouds etc... . Overall the `AircraftVerse` dataset contains multiple modalities and there are many opportunities to test new models out. We take some time in the paper to highlight this.

**Is there anything about the composition of the dataset or the way it was collected and preprocessed/cleaned/labeled that might impact future uses?**  The fidelity of the scientific models are the main factor for users of this dataset to consider when using this data set for future applications.

**Are there tasks for which the dataset should not be used?**  Not that the authors are aware of.

## A.6 Distribution

**Will the dataset be distributed to third parties outside of the entity on behalf of which the dataset was created?**  Yes, the dataset is publicly available.

**How will the dataset will be distributed?**  The dataset is hosted at https://zenodo.org/record/6525446, baseline models and code are at https://github.com/SRI-CSL/AircraftVerse, and the dataset description is at https://aircraftverse.onrender.com/.

**When will the dataset be distributed?**  The dataset was first released in 2023.

**Will the dataset be distributed under a copyright or other intellectual property (IP) license, and/or under applicable terms of use (ToU)?**  See Section B.

**Have any third parties imposed IP-based or other restrictions on the data associated with the instances?**  No.

## A.7 Maintenance

**Who will be supporting/hosting/maintaining the dataset?**  The Neuro-Symbolic Computing and Intelligence (NuSCI) Research Group at SRI International will support and maintain the dataset.

**How can the owner/curator/manager of the dataset be contacted?** Please see the email addresses at the start of the paper.

**Will the dataset be updated?** We intend to continue to add to the dataset as we evaluate further designs.

**If others want to extend/augment/build on/contribute to the dataset, is there a mechanism for them to do so?** Please contact the authors in regards to ways that the dataset could be extended. For example running different scientific models may be possible.

## B   Licensing description

Our dataset uses a Creative Commons Attribution-ShareAlike (CC BY-SA) license. This means that the data can be shared and/or adapted for any purpose under the condition that appropriate credit is given, and that a link to the license is provided, and that users indicate if any changes were made. For more details on the license, please refer to https://creativecommons.org/licenses/by/4.0/.

## C   Flight dynamics model

In this section, we describe the physics of the flight dynamics model (FDM). The vehicle is equipped with five major components 1) propellers, 2) motors, 3) batteries as power source, 4) wings and stabilizers, 5) body structure such as a fuselage or a board where components are connected to. The FDM considers a six-degrees of freedom where the state of the vehicle at time $t$ is defined as follows

$$\boldsymbol{x}^t = (U, V, W, P, Q, R, \boldsymbol{q}, x, y, z, \Omega_i, Q_j). \tag{1}$$

- $(U, V, W)$ is the velocity of the vehicle in the body frame along $X$, $Y$, and $Z$ axis respectively.
- $(P, Q, R)$ is the rotation of the body, i.e., representing roll, pitch, and yaw respectively.
- $\boldsymbol{q}$ is a four-dimensional vector connecting the world frame to body frame.
- $(x, y, z)$ is the location in the world frame.
- $\Omega_i$ is the rotation for the $i$th motor expressed as radian/sec.
- $Q_j$ is the power drawn from $j$th battery.

The flight dynamics is governed by the following equation

$$\frac{d\boldsymbol{x}}{dt} = f(\boldsymbol{x}, \boldsymbol{u}), \tag{2}$$

where $\boldsymbol{x}$ is the current state of the vehicle and $\boldsymbol{u}$ is a vector representing control signals. Each component can be controlled independently by the corresponding control signal and we assume the control signals are within range of 0 and 1. The function $f()$ represents the computation of forces. The computation of various forces as described below.

**Gravity.** Let us consider the velocity at the world frame is $v$ and the body frame is $V$, respectively. The translation matrix connective $v$ to $V$ is defined as $R(\boldsymbol{q})$. Thus, $V = R(\boldsymbol{q})v$. The gravitation force is applied on the the center of mass of the vehicle and computed as

$$F_g = mgR(\boldsymbol{q})\hat{z}, \tag{3}$$

where $\hat{z} = (0, 0, 1)$ is the unit vector pointing down in the world frame.

**Drag.** The drag force on the vehicle's body is computed based on the vehicle's flight through the air. We assume that the velocity is the vehicle's velocity in the air. Thus, the velocity is updated from the body frame to account for the motion of the air being in the world frame as $U - R(\boldsymbol{q})v_w$, where $v_w$ is the wind velocity. Then the body drag is approximated as

$$F_b = -\frac{1}{2}\rho \begin{pmatrix} X_{body,uu}|U + 2v_i|(U + 2U_{in}) \\ Y_{body,vv}|V + 2V_{in}|(V + 2V_{in}) \\ Z_{body,ww}|W + 2W_{in}|(W + 2W_{in}) \end{pmatrix} \tag{4}$$

where $X_{body,uu}, Y_{body,vv}, Z_{body,ww}$ are the constants for each coordinate axis and for the vehicle we set it equal to a constant times the presented area of the vehicle's body along the coordinate axis. $U_{in}, V_{in}, W_{in}$ are the wind velocity along X, Y, and Z directions, respectively. As for the asymmetrical vehicles, center of drag may not align with the center of mass, this generates a moment force given as $M = (X_{cbody} - X_{cm}) \times F_b$, where $X_{cbody}$ is the center of body force and $X_{cm}$ center of mass.

**Propeller and motor modeling.** We define the geometric center of a propeller in the body frame is $X_p$ and the unit normal vector to the propeller shaft is $n_p$. The velocity of the air passing through the stationary propeller is estimtaed as

$$V_p = (V + \vec{\Omega} \times (X_p - X_{cm})) \cdot n_p. \tag{5}$$

For propeller performance, we refer to pre-computed tables for each propeller providing performance data with respect to spin rate and advance ratio $J$. The spin rate is defined by the rotations per second and denoted by $n = \Omega/2\pi$. The advanced ratio is computed as $J = V_p/(nD)$. where $D$ is the diameter of the propeller. The thrust and power of the propeller are estimated as

$$F_p = C_T(n, J)\rho n^2 D^4, P = C_p(n, J)\rho n^3 D^5, \tag{6}$$

where $F_p$ and $P$ are thrust and power. $C_T()$ and $C_p()$ are pre-computed performance values depending on $n$ and $J$. The total force from the propeller is computed by summing over all propellers.

The torque from the motor is estimated as $\tau_{motor} = K_T(I - I_0)$, where $I$ is the current, $I_0$ is the idle current, and $K_T$ is the torque constant. The relation between the current and voltage is given by

$$V_{motor} = V_{battery}u_c = \frac{\Omega}{K_V} + IR_w \tag{7}$$

where $V_{motor}$ is the voltage applied to the motor, $V_{battery}$ is the battery voltage, and $u_c = [0, 1]$ is the control signal driving the motor. $K_V$ is the RPM constant and $R_w$ is the resistance.

Note that the flight dynamics simulation model is currently under construction, and there are some assumptions that would affect the flight characteristics and performance of a given design outside of this particular FDM. For example, the parametric connectors are currently made of 3d ABS plastic which accurately reflects the mass properties of such a material, but may not hold up to the structural loads experienced during flight, which are not currently modeled. Further development will continue to address these simulation gaps.

## D  Additional experiment details

For the baseline experiments listed in this paper, we randomly shuffle the entire dataset and then split the data into train, validation, and test splits of 70 %, 10 %, and 20 % respectively. For the regression tasks, we normalize the data using the mean and standard deviation of the data. Where necessary we remove anomalous data points. For the mass prediction, we remove designs with a mass greater than 35 Kg from the data set. When we report the results in [1] for MSE, we use the unnormalized values.

### D.1  LSTM architecture

The LSTM baseline model consists of a linear input layer followed by an LSTM layer (nn.LSTM in PyTorch) with an input size of 512, a hidden size of 512. The LSTM layer is then connected to a final linear layer, via a dropout layer with a value of 0.5, that takes in all the outputs of the LSTM layer and provides the one-dimensional prediction. During training we use the PyTorch in-built SGD optimizer with a learning rate of 0.05 and set the number of epochs to 4000, where we save the model with the best validation loss.

### D.2  Transformer encoder architecture

The transformer encoder consists of a linear encoding layer followed by a positional encoding layer. The output of the positional encoding is then passed through a transformer encoding layer (nn.TransformerEncoder in PyTorch) with 8 layers and 2 heads. The final token from the transformer encoder is then passed through a linear layer that goes from the embedding dimension of 200 to a single output dimension. During training we use the PyTorch in-built SGD optimizer with a learning rate of 0.1 and set the number of epochs to 4000, where we save the model with the best validation loss.

### D.3 Graph convolutional neural network

Apart from the aforementioned models on sequence data, we also consider a model that directly operates on the point cloud. We represent the point clouds as a set of 3D points and apply a graph convolutional network (GCN) [17, 32] to make predictions on them. We build the graphs with the points as nodes where nodes are represented by their 3D coordinate. Each node is connected to $K$ nearest neighbors ($K$=40). Note that the GCN model captures the structural information from the point clouds and does not have access to the specification of the components. Our GCN has four convolutional layers followed by a fully connected layer. We train the GCN with Adam optimizer for 200 epochs.

## E    Automated data generation

As part of automating our data generation pipeline for `AircraftVerse`, we defined a design grammar. Building a grammar to define a CPS requires significant domain expertise and `AircraftVerse` was no exception. Our focus was to build a procedural generator of designs that produced topologically valid aircraft designs, i.e. designs that met the requirements needed to be evaluated by the scientific models. Appendix G shows an example of a valid design tree. Once we built a valid procedural generator (with expert-guided distributions over all choices), we added the following heuristics: limit designs to having a maximum of 12 wings; limit designs to having a maximum of 16 propellers; set the probability of a recursive structure to be 0.25, such that we do not get infinitely long recursive structures. We then generated half our dataset using our procedural generator. Thereafter, we trained a transformer-based classifier to identify designs from their sequences that were extremely unlikely to hover (similar to the benchmark outlined in Section D). The next stage of generation then used this classifier as a weak filter (calibrated according to a recall of 90 %) to increase the number of hovering vehicles in the dataset. Components of this process are highlighted in our previous technical report [6]. Overall this process is an accumulation of over two years of work in building the right domain knowledge and compute infrastructure to ensure that the data remains diverse, interesting, and useful.

## F    Aircraft grammar and components list

The full JSON schema for the aircraft designs in our `design_tree.json` files is available at https: //github.com/SRI-CSL/AircraftVerse/blob/main/schema/uav_schema.json. As mentioned previously, the sequentialized designs in `design_seq.json` files are pre-order traversals of the design trees (they can be viewed as sequences of key-value pairs).

Below, we also include the list of allowed values for each property (or "key") in the trees/sequences. Note that the JSON schema does not specify the allowed values for the component selection properties like `propType`, `motorType`, etc, as the specific lists of components are subject to change, and not considered part of the design language itself. We list all the currently used values for those properties below.

Table 2: Key-value descriptions.

| KEY | DESCRIPTION | VALUE OPTIONS |
|---|---|---|
| node_type | Main structural components of aircraft, like the central hub and connector arms. | 'AngledPropArm', 'AngledWingArm', 'BendSegment', 'BranchSegment_Asym', 'BranchWithTopSegment_Asym', 'ConnectedHub2_Sym_Wide', 'ConnectedHub3_2_1', 'ConnectedHub4_2_2', 'ConnectedHub4_Sym', 'ConnectedHub6_1_2_2_1', 'ConnectedHub6_2_2_2', 'ConnectedHub6_Sym', 'DoubleBendSegment', 'PropArm', 'SidewaysBendSegment', 'SidewaysBendWithTopSegment', 'WingArm'. |

| KEY | DESCRIPTION | VALUE OPTIONS |
| --- | --- | --- |
| propType | Propeller type. | 'apc_propellers_10x5', 'apc_propellers_10x6', 'apc_propellers_10x7', 'apc_propellers_11x5_5', 'apc_propellers_11x8', 'apc_propellers_12x12', 'apc_propellers_12x6', 'apc_propellers_12x8', 'apc_propellers_13x10', 'apc_propellers_13x4', 'apc_propellers_13x5_5', 'apc_propellers_13x6_5', 'apc_propellers_13x8', 'apc_propellers_14x7', 'apc_propellers_14x8_5', 'apc_propellers_15x10', 'apc_propellers_15x4', 'apc_propellers_16x10', 'apc_propellers_16x4', 'apc_propellers_18x10', 'apc_propellers_18x8', 'apc_propellers_19x10', 'apc_propellers_20x10', 'apc_propellers_4_1x4_1', 'apc_propellers_4_75x4_75', 'apc_propellers_5_5x4_5', 'apc_propellers_5x3', 'apc_propellers_5x5', 'apc_propellers_5x7_5', 'apc_propellers_6x4', 'apc_propellers_6x6', 'apc_propellers_7x4', 'apc_propellers_7x5', 'apc_propellers_7x6', 'apc_propellers_8x6', 'apc_propellers_8x8', 'apc_propellers_9x4_5', 'apc_propellers_9x6' |
| motorType | Motor type. | 'kde_direct_KDE2306XF2550', 'kde_direct_KDE2315XF885', 'kde_direct_KDE2315XF965', 'kde_direct_KDE2814XF_515', 'kde_direct_KDE2814XF_775', 'kde_direct_KDE3510XF_475', 'kde_direct_KDE3510XF_715', 'kde_direct_KDE3520XF_400', 'kde_direct_KDE4012XF_400', 'kde_direct_KDE4014XF_380', 'kde_direct_KDE4213XF_360', 't_motor_AS2308KV1450', 't_motor_AS2308KV2600', 't_motor_AS2312KV1150', 't_motor_AS2312KV1400', 't_motor_AS2317KV1250', 't_motor_AS2317KV1400', 't_motor_AS2317KV880', 't_motor_AS2814KV1050', 't_motor_AS2814KV1200', 't_motor_AS2814KV2000', 't_motor_AS2814KV900', 't_motor_AS2820KV1050', 't_motor_AS2820KV1250', 't_motor_AS2820KV880', 't_motor_AT2308KV1450', 't_motor_AT2308KV2600', 't_motor_AT2310KV2200', 't_motor_AT2312KV1150', 't_motor_AT2312KV1400', 't_motor_AT2317KV1250', 't_motor_AT2317KV1400', 't_motor_AT2317KV880', 't_motor_AT2321KV1250', 't_motor_AT2321KV950', 't_motor_AT2814KV1050', 't_motor_AT2814KV1200', 't_motor_AT2814KV900', 't_motor_AT2820KV1050', 't_motor_AT2820KV1250', 't_motor_AT2820KV880', 't_motor_AT2826KV900', 't_motor_AT3520KV550', 't_motor_AT3520KV720', 't_motor_AT3520KV850', 't_motor_AT3530KV580', 't_motor_AT4120KV500', 't_motor_AT4120KV560', 't_motor_AT4125KV540', 't_motor_AT4130KV450', 't_motor_MN2212KV780', 't_motor_MN2212KV920', 't_motor_MN3110KV470', 't_motor_MN3110KV700', 't_motor_MN3110KV780', 't_motor_MN3508KV380', 't_motor_MN3508KV580', 't_motor_MN3508KV700', 't_motor_MN3510KV360', 't_motor_MN3510KV630', 't_motor_MN3510KV700', 't_motor_MN3515KV400', 't_motor_MN3520KV400', 't_motor_MN4010KV370', 't_motor_MN4010KV475', 't_motor_MN4010KV580', 't_motor_MN4012KV340', 't_motor_MN4012KV400', 't_motor_MN4012KV480', 't_motor_MN4014KV330', 't_motor_MN4014KV400', 't_motor_MN5208KV340', 't_motor_MN5212KV340', 't_motor_MN5212KV420', 't_motor_MT22081100KV', 't_motor_MT2216V2800KV' |
| batteryType | Battery type. | 'TurnigyGraphene1000mAh2S75C', 'TurnigyGraphene1000mAh3S75C', 'TurnigyGraphene1000mAh4S75C', 'TurnigyGraphene1000mAh6S75C', 'TurnigyGraphene1200mAh6S75C', 'TurnigyGraphene1300mAh3S75C', 'TurnigyGraphene1300mAh4S75C', 'TurnigyGraphene1400mAh3S75C', 'TurnigyGraphene1400mAh4S75C', 'TurnigyGraphene1500mAh3S75C', 'TurnigyGraphene1500mAh4S75C', 'TurnigyGraphene1600mAh4S75C', 'TurnigyGraphene1600mAh4S75CSquare', 'TurnigyGraphene2200mAh3S75C', 'TurnigyGraphene2200mAh4S75C', 'TurnigyGraphene3000mAh3S75C', 'TurnigyGraphene3000mAh5S75C', 'TurnigyGraphene3000mAh6S75C', 'TurnigyGraphene4000mAh3S75C', 'TurnigyGraphene4000mAh4S75C', 'TurnigyGraphene4000mAh6S75C', 'TurnigyGraphene5000mAh3S75C', 'TurnigyGraphene5000mAh4S75C', 'TurnigyGraphene5000mAh6S75C', 'TurnigyGraphene6000mAh3S75C', 'TurnigyGraphene6000mAh4S75C', 'TurnigyGraphene6000mAh6S75C' |
| wingType | Wing type. | 'NACA_0012', 'NACA_0015', 'NACA_0018', 'NACA_0021', 'NACA_0025', 'NACA_2212', 'NACA_2306', 'NACA_2309', 'NACA_2312', 'NACA_2315', 'NACA_2406', 'NACA_2409', 'NACA_2412', 'NACA_2415', 'NACA_2418', 'NACA_2421', 'NACA_2506', 'NACA_2509', 'NACA_2512', 'NACA_2515', 'NACA_2518', 'NACA_2521', 'NACA_2612', 'NACA_2712', 'NACA_4212', 'NACA_4306', 'NACA_4309', 'NACA_4312', 'NACA_4315', 'NACA_4318', 'NACA_4321', 'NACA_4406', 'NACA_4409', 'NACA_4412', 'NACA_4415', 'NACA_4418', 'NACA_4421', 'NACA_4506', 'NACA_4509', 'NACA_4512', 'NACA_4515', 'NACA_4518', 'NACA_4521', 'NACA_4612', 'NACA_4712', 'NACA_6212', 'NACA_6306', 'NACA_6309', 'NACA_6312', 'NACA_6315', 'NACA_6318', 'NACA_6321', 'NACA_6406', 'NACA_6409', 'NACA_6412', 'NACA_6415', 'NACA_6418', 'NACA_6421', 'NACA_6506', 'NACA_6509', 'NACA_6512', 'NACA_6515', 'NACA_6518', 'NACA_6521', 'NACA_6612', 'NACA_6712' |
| servoType | Servo type for control of flaps and ailerons. | 'Hitec_D485HW', 'Hitec_D89MW', 'Hitec_D954SW', 'Hitec_HS_225BB', 'Hitec_HS_225MG', 'Hitec_HS_40', 'Hitec_HS_45HB', 'Hitec_HS_5055MG', 'Hitec_HS_5065MG', 'Hitec_HS_5070MH', 'Hitec_HS_5087MH', 'Hitec_HS_5245MG', 'Hitec_HS_53', 'Hitec_HS_5496MH', 'Hitec_HS_55', 'Hitec_HS_5565MH', 'Hitec_HS_625MG', 'Hitec_HS_645MG', 'Hitec_HS_65HB', 'Hitec_HS_65MG', 'Hitec_HS_70MG', 'Hitec_HS_7235MH', 'Hitec_HS_7245MH', 'Hitec_HS_81', 'Hitec_HS_82MG', 'Hitec_HS_85BB', 'Hitec_HS_85MG' |
| wingRot | The wing's angle. | Float: [0, 360] |
| chord | The wing's chord. | Float: [100, 500] |
| span | The wing's span. | Float: [200, 2000] |

| KEY | DESCRIPTION | VALUE OPTIONS |
|---|---|---|
| armLength | A context specific arm length generally associated with node_types such as 'PropArm' and 'WingArm'. | Float: [50, 500] |
| arm1Length | A more specific arm length associated with arms that include a bend. | Float: [50, 500] |
| arm2Length | A more specific arm length associated with arms that include a bend. | Float: [50, 200] |
| angle | A context specific angle that corresponds to angles in the vehicle construction. | Float: [-135, 360] |
| offset | A context specific offset value that can correspond to structural components such as the wing offset and the flange offset. | Float: [-200, 90] |
| x1_offset | Offset in the x-direction of Battery on the central plate. | Float: [0, 120] |
| z1_offset | Offset in the z-direction of Battery on the central plate. | Float: [0, 120] |
| tube_offset | The wing tube offset. | Float: [10, 90] |

# G   Aircraft design summary

In the additional supplementary material, we include the complete representation of an individual design when accessed from `AircraftVerse`, where we print the:

- `design_tree.json`
- `design_seq.json`
- `cadfile.stl`
- `pointCloud.npy`
- `output.json`
- `trims.npy`

As part of the notebook, we show how to access the design corpus and provide an example of printing the specification of a battery. Finally, we also include an enlarged plot of the trim states for readability, as Figure 2 was an illustration of a full design that by necessity limited the allocated space for it in the full paper.

