# DataDemo-Paper

May 26, 2023

```python
[1]: import os
     import json
     import pickle
     import numpy as np
     import sys
     sys.path.append("./code")
     import util
     import matplotlib.pyplot as plt
     %matplotlib inline

     path = '<dataset path>'
     corpus = '<Corpus Path>'
     design = 'design_n'
```

**Show data folder structure**

```python
[2]: data_files = os.listdir(os.path.join(path,design))
     print(data_files)
```

```
['design_seq.json', 'design_tree.json', 'design_low_level.json', 'Geom.stp',
'cadfile.stl', 'trims.npy', 'output.json', 'pointCloud.npy']
```

**Show design tree**

```python
[3]: tree = open(os.path.join(os.path.join(path,design), 'design_tree.json'))
     tree_json = json.load(tree)
     tree_json_print = json.dumps(tree_json, indent=4)
     tree.close()
     print(tree_json_print)
```

```
{
    "generator_version": "UAV2_gen12",
    "name": "design_12028_e437c0c4835841a99d48808aa8734bc7_s",
    "hub": {
        "node_type": "ConnectedHub4_Sym",
        "mainSegment": {
            "node_type": "PropArm",
            "armLength": 413.5996130739701,
            "motor": {
```

```
                    "motorType": "t_motor_AT3530KV580"
                },
                "prop": {
                    "propType": "apc_propellers_18x8"
                },
                "flange": {
                    "offset": 41.65045980035022,
                    "angle": 0.0
                }
            }
        },
        "fuselageWithComponents": {
            "node_type": "SingleBatteryFuselageWithComponents",
            "battery": {
                "batteryType": "TurnigyGraphene2200mAh3S75C"
            },
            "fuselage": {
                "length": 106.0,
                "vertDiameter": 52.5,
                "horzDiameter": 332.4,
                "floorHeight": 8.75,
                "batteryX": 0.0,
                "batteryY": 6.0,
                "rpmX": 0.0,
                "rpmY": -100.8,
                "autoPilotX": 0.0,
                "autoPilotY": 44.5,
                "currentX": 0.0,
                "currentY": -80.65,
                "voltageX": 0.0,
                "voltageY": 98.3,
                "gpsX": 0.0,
                "gpsY": -40.5,
                "varioX": 0.0,
                "varioY": 76.65
            }
        }
    }
```

**Design Corpus**

- One can use the design corpus to add context to the component types. For example, here is a print out of the attributes of a single battery.

```
[4]: corpus_file = open(corpus, "rb")
     corpus_dic = pickle.load(corpus_file)
     corpus_file.close()
     print(corpus_dic.keys())
```

```
print('\nBattery Example:\n')
print('Number of batteries: ', len(corpus_dic['Battery'].keys()))
print('Battery example: TurnigyGraphene2200mAh3S75C:\n')
for key, value in corpus_dic['Battery']['TurnigyGraphene2200mAh3S75C'].items():
    print('{:20s}: {:}'.format(key, value))
```

```
dict_keys(['Servo', 'GPS', 'ESC', 'Wing', 'Sensor', 'Propeller', 'Receiver',
'Motor', 'Battery', 'Autopilot'])

Battery Example:

Number of batteries:  36
Battery example: TurnigyGraphene2200mAh3S75C:

CAPACITY            : 2200.0
CONT_DISCHARGE_RATE : 75.0
COST                : 32.73
LENGTH              : 106.0
PEAK_DISCHARGE_RATE : 150.0
THICKNESS           : 31.0
VOLTAGE             : 11.1
WEIGHT              : 0.23
WIDTH               : 35.0
```

**Design Sequence**

- As described in the paper, we convert the design tree into a design sequence which we find to be useful for sequence based machine learning approaches.
- Here we show an example sequence that corresponds to the tree above.

```
[5]: seq = open(os.path.join(os.path.join(path,design), 'design_seq.json'))
     seq_json = json.load(seq)
     seq_json_print = json.dumps(seq_json, indent=4)
     seq.close()
     print(seq_json_print)
```

```
[
    {
        "generator_version": "UAV2_gen12"
    },
    {
        "name": "design_12028_e437c0c4835841a99d48808aa8734bc7_s"
    },
    {
        "node_type": "ConnectedHub4_Sym"
    },
    {
        "node_type": "PropArm"
```

```
    },
    {
        "armLength": 413.5996130739701
    },
    {

        "motorType": "t_motor_AT3530KV580"
    },
    {

        "propType": "apc_propellers_18x8"
    },
    {

        "offset": 41.65045980035022
    },
    {

        "angle": 0.0
    },
    {

        "node_type": "SingleBatteryFuselageWithComponents"
    },
    {

        "batteryType": "TurnigyGraphene2200mAh3S75C"
    },
    {

        "length": 106.0
    },
    {

        "vertDiameter": 52.5
    },
    {

        "horzDiameter": 332.4
    },
    {

        "floorHeight": 8.75
    },
    {

        "batteryX": 0.0
    },
    {

        "batteryY": 6.0
    },
    {

        "rpmX": 0.0
    },
    {

        "rpmY": -100.8
    },
    {

        "autoPilotX": 0.0
```

```
        },
        {
            "autoPilotY": 44.5
        },
        {
            "currentX": 0.0
        },
        {
            "currentY": -80.65
        },
        {
            "voltageX": 0.0
        },
        {
            "voltageY": 98.3
        },
        {
            "gpsX": 0.0
        },
        {
            "gpsY": -40.5
        },
        {
            "varioX": 0.0
        },
        {
            "varioY": 76.65
        }
    ]
```

### 0.0.1 STL plot

```
[6]: util.plot_stl(path, design)
```

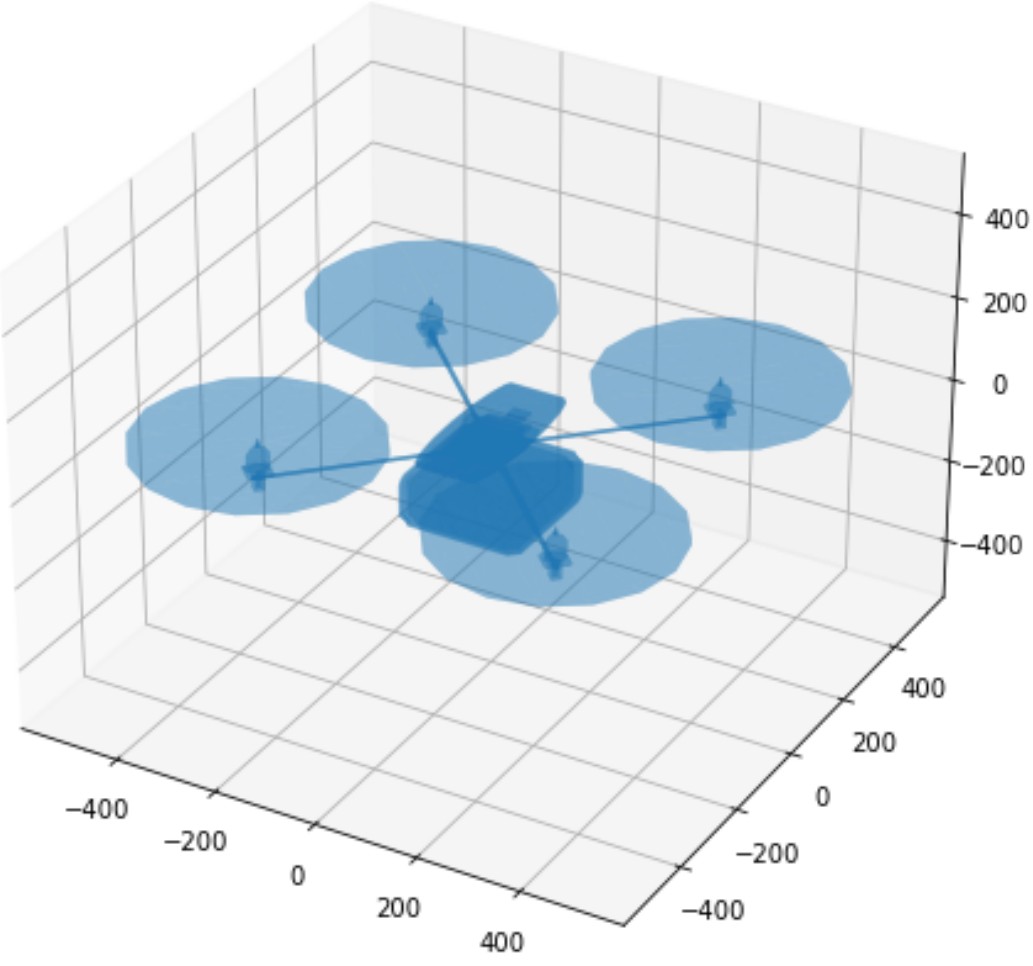

### 0.0.2 Point Cloud

- One can easily convert the STL file into a point cloud... Here is one we made earlier and have included in the data set.

```
[7]: util.plot_pointCloud(path, design)
```

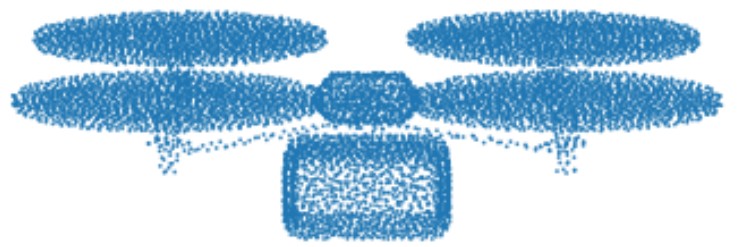

### 0.0.3 Output File

```
[8]: seq = open(os.path.join(os.path.join(path,design), 'output.json'))
     seq_json = json.load(seq)
     seq_json_print = json.dumps(seq_json, indent=4)
     seq.close()
     print(seq_json_print)
```

```
{
    "Interferences": 2,
    "Mass": 3.4330805367769828,
    "Batt_amps_ratio_MFD": 0.72659409,
    "Batt_amps_ratio_MxSpd": 0.906760275,
```

```
      "Distance_MxSpd": 1228.10852,
      "Max_Distance": 1268.38354,
      "Hover_Time": 83.8901367,
      "Max_Speed": 29.0,
      "Max_uc_at_MFD": 0.738918006,
      "Mot_amps_ratio_MFD": 0.602965593,
      "Mot_amps_ratio_MxSpd": 0.807905734,
      "Mot_power_ratio_MFD": 0.310742617,
      "Mot_power_ratio_MxSpd": 0.416359991,
      "Power_MFD": 872.750122,
      "Power_MxSpd": 1350.86023,
      "Speed_MFD": 24.0
  }
```

### 0.0.4 Plot the trims for the UAV design

```python
[9]: trims = np.load(os.path.join(os.path.join(path,design),'trims.npy'))
     names = ['distance',
      'flight_time',
      'pitch_angle',
      'max_uc',
      'thrust',
      'lift',
      'drag',
      'current',
      'total_power',
      'frac_amp',
      'frac_pow',
      'frac_current']
     plt = util.plot_trim_stats(trims, names, fs = 19, figsize = (23,2.))
     plt.show()
```

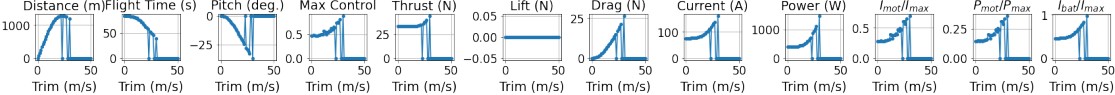

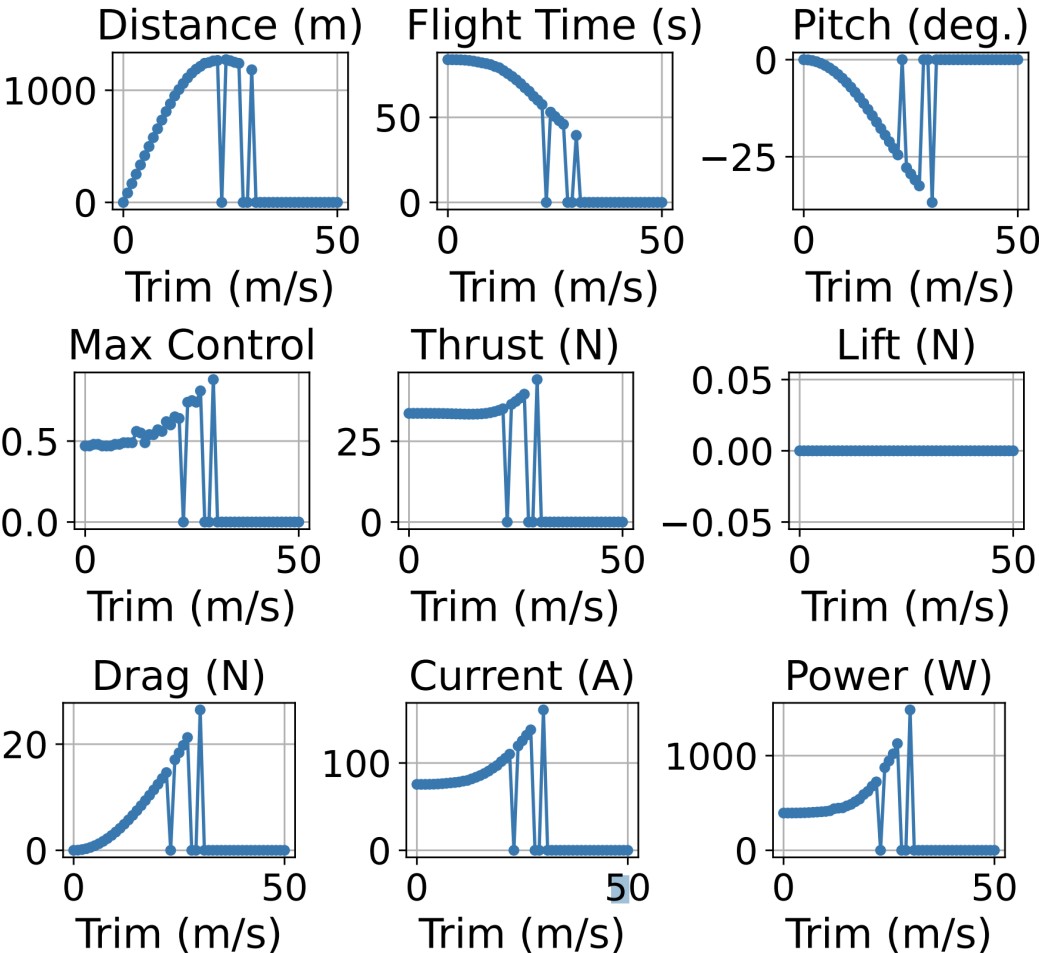

Figure 7: Analysis of trims corresponding to different velocities from 0 to 50 m/s