# OpenReview forum: "AircraftVerse: A Large-Scale Multimodal Dataset of Aerial Vehicle Designs"
_NeurIPS.cc/2023/Track/Datasets_and_Benchmarks — NeurIPS 2023 Datasets and Benchmarks Poster_

### Official Review · Reviewer_29GD · 2023-07-21
**Review of AircraftVerse: A Large-Scale Multimodal Dataset of Aerial Vehicle Designs**

**Rating:** 9
**Confidence:** 2

**Strengths:**

### Main strengths
#### A novel multi-modal dataset pioneering the field
The authors addressed an existing gap in the domain of aircraft designs, which seems well grounded and seriously developed. This is clearly the principal contribution of this work.
#### Aiming to address a gap in multi-modal holistic engineering design
There were no available corpus of such diverse aircraft designs accompanied by their design details and also the results from dynamic flight simulations and/or additional engineering evaluations.
#### Performance Metrics.
The dataset includes evaluation results from high fidelity physics models (both custom flight dynamic simulators and commercial, i.e. Creo), enabling researchers to explore and analyze performance metrics such as maximum flight distance and hover-time for each design.


**Additional Feedback:**

Overall, the paper and the technical work demonstrate satisfactory quality. However, considering the nature of the dataset and its potential implications for harmful applications, I strongly advise subjecting it to a specialized ethics review. It is crucial to carefully evaluate the ethical aspects and potential risks associated with the dataset.

**Clarity:**

### Paper clarity.
The paper is generally well written.


**Correctness:**

### Paper claims
Based on my current knowledge, I believe that the main claims by the authors are accurate.
### Dataset construction
The construction of the dataset appears to have been conducted with fairness and thoroughness. Nonetheless, it is worth noting that certain minor details are absent, such as a comprehensive description of the interactions with experts and how these interactions influenced their procedural modeling process. Including such information would offer a more comprehensive understanding of the dataset's development and potentially strengthen its overall validity and usefulness.
### Benchmark and evaluation
Based on the comprehensive analysis provided in the previous text, the authors have made a sound and well-reasoned decision by incorporating both custom flight dynamic simulators and commercial simulation software (Creo) for evaluation purposes. This approach adds credibility to their research and enhances the validity of their findings. Moreover, the authors have effectively demonstrated the multi-modal nature of the dataset by presenting various representations of the aircraft designs, and they have gone the extra mile by providing all the simulation results. This availability of detailed simulation outcomes enables a thorough assessment of the predicted performance of the proposed designs, bolstering the reliability and significance of their contributions.


**Documentation:**

I believe yes.


**Ethics:**

#### Responsible Use.
The dataset's potential applications for aircraft vehicles need to be considered carefully from an ethical perspective. Researchers should be aware of how the data might be used and ensure that its application aligns with ethical principles, avoiding any harmful practices.


**Limitations:**

I think authors provided a fair description of their dataset and potential imitations, given its experimental nature.


**Opportunities For Improvement:**

#### How the search was performed? How diversity was ensured?
Authors mentioned in l85 that their search for designs included over a hundred thousand candidates. How that search was developed? was it done manually, or machine-supervised? I would be curious to know how that task was resolved. Also authors claim a diversity-preserving choice. Authors explained that they also relied on experts heuristics, how one can ensure diversity is preserved, what if your experts neglected a specific design that could be successful? Just curious to know if they have any comment ono that.
#### Baseline surrogate models.
AUthors provide a very high-level explanation of how the transformer encoder model processes their symbolic representations. Perhaps a more detailed explanation on that aspect would be clarifying, since at the end it seems to be the machine learning model proposed to assess the proposed designs.
#### Procedural designs and expert heuristics
In this case it would be interesting to know how did the authors integrate heuristics into a procedural modeling approach. A bit more details would be welcome, since this is an very interesting challenge (how to not include undesired biased decisions into the procedural modeler). Would you have any strategy detecting that issue “Might we have neglected a certain space of all the possible designs due to the experts' biases”?
#### Baseline models.
While many details are provided in the Appendix D., when discussing the transformer encoder and the LSTM at least a high level description of how those networks work in essence in conjunction with this dataset, and how they were trained and tested would be appropriate to appear in the main paper.
#### Other comments:
##### Minor typo.
l290 informative?


**Relation To Prior Work:**

### Related work.
Up to my knowledge, the related work was fairly discussed.


**Summary And Contributions:**

### Summary
This submission presents a new publicly available dataset called "AircraftVerse," which focuses on aerial vehicle design. The proposed dataset contains > 25k diverse air vehicle designs, making it the largest corpus of engineering designs with such complexity. Each design in the dataset is represented through multiple modalities, encompassing different physics domains, including symbolic design trees, STandard for the Exchange of Product (STEP) model data, 3D CAD designs in stereolithography (STL) format, 3D point clouds representing the design's shape, and evaluation results from high fidelity state-of-the-art physics models that measure performance metrics like maximum flight distance and hover-time.
### Contributions
The main contributions of this work are:
#### Dataset
The "AircraftVerse" dataset provides a rich collection of 27,714 aircraft designs, each with associated metadata summarizing the design's performance.
#### Multi-Modal Representation: The dataset introduces a multi-modal approach to represent aircraft designs, incorporating various physics domains, and providing a holistic view of the design process.
#### Baseline Surrogate Models: The authors developed baselines of surrogate models that use different modalities of design representation to predict design performance metrics. These models serve as a starting point for researchers to delve into machine learning applications within aircraft design and cyber-physical systems (CPS).

#### Potential Impact: The authors discuss the potential impact of the "AircraftVerse" dataset on the use of learning in aircraft design and more broadly in CPS. By providing access to this extensive dataset, researchers can explore various interdisciplinary applications of machine learning in aircraft design and related fields.
In summary, this reviewer believes that the "AircraftVerse" dataset and the associated experiments open up new possibilities for researchers to explore and address different challenges and approaches in the realm of aircraft design and cyber-physical systems using machine learning techniques.

---

> ### Author Response · Authors · 2023-08-14
> **Response to 29GD**
>
> We thank the reviewer for their very detailed and positive review. We now aim to answer specific questions in addition to the more general response in the summary at the top.
>
> > Just curious to know if they have any comment on that. /Procedural designs and expert heuristics
>
> Thanks for the important question. We have now released the probabilistic generator that we used to sample designs. Incorporating domain-knowledge at the expense of diversity was a key challenge. The probabilistic generator enforces certain symmetries since this was a common heuristic we learned while working with the domain experts. It is now possible to see from the generator, the distributions over sampling different components are made as general as possible which leads to the diverse set of designs. One example of this can be seen in the code: https://github.com/SRI-CSL/AircraftVerse/blob/811e3f254c7ef3236fdb688fc81575e0b156d478/prob_gen/src/main/scala/com/sri/nscore/uav2/DefaultUAV2Generator.scala#L29. This line shows how we favor sampling wing/prop arms over sampling additional connectors following the categorical distribution with probabilities [0.7, 0.3]. We found that this balanced the possibility of sampling diverse designs with the possibility of getting infinitely recurring connectors.
>
> We have now implemented your feedback regarding incorporating an ethics statement. Thanks once again for the detailed review. We have also fixed the typo.

---

> > ### Comment · Reviewer_29GD · 2023-08-25
> > **The submission remains technically sound. However, it remains critical to address important ethical concerns.**
> >
> > I would like to thank the authors for providing response to my review comments.
> >
> > My understanding of how experts knowledge was integrated with procedural generation - given the code example - is that basically they embedded this expertise into the code itself, weighting the sampling of expert rules versus procedural designs using some probabilistic weight, but in the end it seems sort of hard-coded knowledge in the system. Spending a line explaining the high-level idea behind it may greatly help readers better understand it.
> >
> > Since I have already expressed my positive impression about the technical side of this work I have not much to add on that front.
> >
> > In my humble opinion, right now the ethical concerns raised by the expert reviewers in ethics, seems the most critical issue to address from now.
> >
> > Kind regards.

---

> > > ### Author Response · Authors · 2023-08-28
> > > **Thanks!**
> > >
> > > Thanks very much for your response!
> > >
> > > Additionally, this comment highlighted that any response to an ethics reviewer is not made visible to all reviewers on OpenReview (which seems to be a flaw). We have highlighted to the ethics reviewers that we have now used our additional page to add a detailed ethics statement. We will see if it is possible to make our responses to the ethics reviewers public.

---

### Official Review · Reviewer_TaAN · 2023-07-22

**Rating:** 5
**Confidence:** 4

**Strengths:**

- Standardizing and indexing tens of thousands of 3D engineering models of aircraft is no small feat, and the authors' efforts in compiling not just geometric descriptors (e.g., shape) but also functional descriptors (e.g., hover time, flight distance) is very helpful for future research efforts.

- AircraftVerse provides data from a non-traditional machine learning domain (aircraft design and functionality).

**Additional Feedback:**

N/A

**Clarity:**

The writing requires another round of editing, it contains numerous grammatical errors such as Line 123: "We use a combination of wings and propellers (in the right topology) provides the thrust and lift needed for efficient horizontal flight and vertical take-off and landing."

**Correctness:**

Dataset construction details are missing from the paper (i.e., details of the aircraft generation model), so it is difficult to comment on the soundness of the process.

**Documentation:**

Dataset construction details are missing from the paper (i.e., details of the aircraft generation model).

**Ethics:**

No.

**Limitations:**

- Besides comments about the fidelity of the flight dynamics model, other dataset limitations were not listed in the paper. To expand this information in the main paper, I could envision that the authors might discuss limitations of their generative design process (e.g., why were so many unusable designs generated that required post-hoc filtering?) or the limitations of characterizing overall aircraft performance through shape files and component performance data. For example, on the scale of simple simulations to real-world full-scale wind tunnels, what is missing from this paper's flight dynamics model to characterize the real-world performance of these designs?


**Opportunities For Improvement:**

- The highest-level opportunity for improvement is making clear in the main paper that each of the designs in AircraftVerse were _probabilistically generated_ and not hand-drawn CAD models collected as part of a multi-year design process for some aerial vehicle by an aerospace entity (for example). I had to dig through the model card to find this information, as the origin of the designs and the process of filtering them is not made very clear in the main body of the text (e.g., the very long paragraph starting at Line 120 does not detail the origins of the data).

    Releasing the outputs from a generative process for downstream use does not seem as useful to the community as releasing the generative model/process itself. What are the details of the probabilistic design generator? Without such details, it is difficult for this review to recommend acceptance for the paper as-is.

- Why would it be useful to predict flight characteristics from 3D point clouds? The aerospace industry has numerous powerful tools that already evaluate performance from CAD files, why are they insufficient for design learning? Is it because they are slow to run? Reading the appendix, the design analyses took 3 minutes to execute, which does not seem onerously long.

- Why would 3D point clouds be used to identify shape interference compared to CAD drawings themselves? Is it just that it enables point-based methods like GNNs to be applied?

- In Figure 6, predicting the ability to hover seems like a simple test, classification of hover time (e.g., in bins of 10 seconds) or directly regressing hover time seems like a more useful output for such models. Were these capabilities evaluated?

**Relation To Prior Work:**

Yes

**Summary And Contributions:**

In this work, a large-scale multimodal dataset of aircraft designs named AircraftVerse is presented. The dataset is comprised of 27,000+ probabilistically-generated CAD models (in STL format), 3D point clouds, and flight characteristics from a high-fidelity simulator. The authors also demonstrate how such a dataset can be used to learn surrogate models that use varying design representation modalities to predict flight characteristics.

---

> ### Author Response · Authors · 2023-08-14
> **Response to TaAN**
>
> We thank the reviewer for their constructive feedback and will highlight how we have changed the work to hopefully meet the raised concerns.
> > What are the details of the probabilistic design generator? Without such details, it is difficult for this review to recommend acceptance for the paper as-is.
>
> We have now released the probabilistic design generator in our github repo with a corresponding README.md describing how to run it. Please see https://github.com/SRI-CSL/AircraftVerse/tree/main/prob_gen. This contains the exact details of how we generate each design. Specifically, https://github.com/SRI-CSL/AircraftVerse/blob/main/prob_gen/src/main/scala/com/sri/nscore/uav2/DefaultUAV2Generator.scala contains the distributions of interest.
>  > Why would it be useful to predict flight characteristics from 3D point clouds?
>
> The purpose of this dataset is to provide the community with a useful tool to explore a new domain and challenge current ML models. Point clouds are a commonly used representation for 3D shapes within the ML community, which is why we chose to provide them in addition to the other modalities. In particular, we do include the CAD files in case future users of the dataset are able to develop ML models that directly operate on this format.
> >  Is it just that it enables point-based methods like GNNs to be applied?
>
> This is an interesting point. In our paper we are mainly using point-clouds to show the utility of other modalities of the dataset for ML models. However, we envision others combining multiple modalities together. Therefore since we are not aware of ML models that can ingest `stl` formats directly, we decided that point clouds were a good intermediate step to enable people to work with the 3D representations.
> > Were these capabilities evaluated?
>
> Table 1 includes regression results for maximum hover time, maximum distance, and mass.

---

### Official Review · Reviewer_v2Uf · 2023-07-30
**Review for Paper 384This paper introduces Aircraftverse, a comprehensive and publicly accessible dataset comprising large-scale aerial vehicle designs. Each design is accompanied by essential metadata to facilitate systematic evaluation of aircraft designs. Furthermore, the paper assesses the dataset's quality using various baseline methods.**

**Rating:** 5
**Confidence:** 3
**Correctness:** Yes
**Clarity:** Yes

**Strengths:**

-This paper addresses the scarcity of large CAD datasets in a specific domain, particularly those incorporating aircraft metadata from FDM examinations.
-The dataset includes a design tree, facilitating aircraft assembly and benefiting evaluation models utilizing compositional assembly (e.g., LEGO assembly paper "Break and make: Interactive structural understanding using lego bricks").
-Furthermore, the paper extensively evaluates the dataset using various baseline methods.

**Additional Feedback:**

No, but if the author can address my concern, i would change my rating.

**Documentation:**

The paper provide dataset and repository publicly available.

**Limitations:**

There isn't a section to cover the limitation of the paper, at least it is not make clear to the reviewer

**Opportunities For Improvement:**

One of the author's primary motivations for creating this dataset, as stated, is to enhance the design of future aircraft. However, during their experiments and evaluations, there is no assessment of the ability to use this dataset to develop innovative aircraft structures. Instead, it focuses only on characterizing existing aircraft designs. Is there a specific reason for this?

If all the aircraft are procedurally generated, why is there a fixed number of designs? Is the pipeline to generate more designs open-source so others can utilize it to produce additional data with new components?

The models seem to predict flight characteristics exceptionally well, with most baselines demonstrating high performance. Therefore, is this dataset valuable for others to use and engage with their proposed models?


**Relation To Prior Work:**

Yes

**Summary And Contributions:**

This paper introduces Aircraftverse, a comprehensive and publicly accessible dataset comprising large-scale aerial vehicle designs. Each design is accompanied by essential metadata to facilitate systematic evaluation of aircraft designs. Furthermore, the paper assesses the dataset's quality using various baseline methods.

---

> ### Author Response · Authors · 2023-08-14
> **Response to v2Uf**
>
> We thank the reviewer for their constructive feedback and will highlight how we have changed the work to hopefully meet the raised concerns.
>
> We have now included a limitations section, as highlighted in our general summary response.
>
> We have also released our procedural generator (or probabilistic generator) as part of our github repo code with a readme to instruct users how to use it. We hope that this helps improve the clarity of how we generated designs. The full pipeline was first published in joint works by Walker et al. 2022 and Bapty et al. 2022. This pipeline requires a CREO licence and additional software to set up. We would suggest contacting these authors if there is interest in setting the pipeline up. We highlight that it is not unusual for domain-specific simulation software to require additional licenses. Overall, we believe that the strengths of making this unique dataset available to the community outweigh the potential challenges in setting up the pipeline.
>
> Thanks for your question on why the focus is on characterizing designs. At this point in time the purpose of this work is to release a unique dataset that contains multiple-modalities that can be leveraged by a range of ML approaches. Therefore it is possible to train appropriate models and test new ideas out on this dataset that may be useful for future work involving aircraft design. We now highlight such limitations in the newly added limitations section.
>
> Please also refer to our highlighted changes that we have made since the reviews and do let us know if they address your concern and enable you to raise your score.

---

> > ### Comment · Reviewer_v2Uf · 2023-08-15
> > **Thank you**
> >
> > I would thank the author for the rebuttal and clearing my doubts, i would take all of these into consideration.

---

### Official Review · Reviewer_vKs1 · 2023-08-01
**Learning-based Aerial Vehicle Design Needs Practical Evidence to Prove Its Effectiveness**

**Rating:** 6
**Confidence:** 2
**Correctness:** The dataset is designed in a right way.

**Strengths:**

(1) This work is quite new for drone design, which can be called an application of 'AI for Engineering'.
(2) The data have been carefully collected and grouped.

**Additional Feedback:**

NA

**Clarity:**

(1) The baseline model should be explained with more details.
(2) Related work is weak.

**Documentation:**

These parts should also be enriched. The website (quite encouraged) is nice but needs more description yet.

**Ethics:**

Its good.

**Limitations:**

(1) The dataset is not available now;
(2) A tutorial is highly expected.
(3) Who are the target users? Drone desiners or machine learning researcher?

**Opportunities For Improvement:**

(1) Related work is too weak to support the dataset as well as the learning-based approach. One may doubt if the scheme really works.
(2) Drone design is a complex job. Are there any successful learning-based cases?

**Relation To Prior Work:**

This part is missed in the manuscript.

**Summary And Contributions:**

This dataset and the associated manuscript have presented the so-called AircraftVerse, a publicly available aerial vehicle design dataset, which contains 27714 diverse air vehicle designs. Each design is associated metadata summarizing the performance of the design. This work may enable researchers to explore a new interdisciplinary area of application for machine learning.

---

> ### Author Response · Authors · 2023-08-14
> **Response to vKs1**
>
> We appreciate the feedback. Please see our response to the highlighted limitations:
> > A tutorial is highly expected.
>
> We have focused on improving the documentation by extending on the README.md to instruct a user how to use the data. We also refer to the three notebooks in the github repo for tutorials with markdown descriptions.
> > The dataset is not available now
>
> The dataset is available at https://zenodo.org/record/6525446 and has been there since submission. Please let us know if you have any issues accessing it.
> > Who are the target users?
>
> We have also added a limitations section that now specifically tackles the query who should be the target audience. Please see the updated paper on OpenReview.
> > Relation to prior work is missing
>
> We also highlight our related work section as to where we cover previous available CAD datasets. We believe our work to be unique in the complexity and scale compared to what is currently available for a CPS dataset.
>
> Please refer to our summary statement at the top for further details of what has changed after review. Let us know if you have any outstanding feedback that can help improve the paper.

---

> > ### Comment · Reviewer_vKs1 · 2023-08-25
> > **Response to the authors for the rebuttal**
> >
> > I would like to thank the authors for the explanations. Nevertheless, there are still concerns including:
> > (1) The website https://zenodo.org/record/6525446 is inaccessible. If the AE or other readers can access it, then neglect this problem please.
> > (2) I am afraid that the newly added contents about the target users or the tutorials are still limited.
> > (3) As for the related work, the authors have not highlighted the significant value of the dataset. The introduction is weak, while the related work is thin. It is recommanded that the authors could tell the readers how this dataset can promote related research, or what the dataset will bring the plane design field.
> > Therefore, I hope the authors can further improve the manuscript and the work.
> > I would like to retain the rating (borderline).
> > Again, thanks to the authors.

---

> > > ### Author Response · Authors · 2023-08-28
> > > **Thanks for Reviewer Response**
> > >
> > > Thanks for the additional feedback. Our response to the above points is as follows:
> > >
> > > > (1) The website https://zenodo.org/record/6525446 is inaccessible
> > >
> > > This website link to Zenodo should work and we have tested the link with multiple people both internal and external to our organizations without any issues. Please consider contacting https://zenodo.org/support if you are having issues. Zenodo is a general-purpose open repository which we have used it before without any issues.
> > >
> > > > (2) I am afraid that the newly added contents about the target users or the tutorials are still limited.
> > >
> > > Please try out these tutorials that we have converted from the GitHub repo to a Colab notebook that we believe to be helpful:
> > > * This is useful for plotting the data: https://colab.research.google.com/drive/1ca4d_89B4ZC-mQ1wrgIjgaY3hmllqdex?usp=sharing
> > > * This is useful for testing out our transformer baseline: https://colab.research.google.com/drive/1Mh7-FPCKD6We4MA-yVd0lqiiOpuntaax?usp=sharing
> > >
> > > There should be enough in these notebooks to help people get accustomed to the dataset. We do not believe these tutorials to be limited.
> > >
> > > We have also tried to write the paper in a way that demonstrates the utility of the data to target users consisting of ML researchers and domain experts. Making this unique and complex CPS dataset freely available to the community, while being as careful as possible to explain and demonstrate both the strengths and limitations of the dataset should make it clear to anyone reading the paper if the dataset would be useful to them. This is what we added to the paper since the review:
> > >
> > > > "... we still highlight the value of this dataset as useful for machine learning researchers looking to explore a novel CPS dataset as well as domain experts looking to explore the potential of data-driven approaches for design. This is precisely what we found to be the case throughout the collaboration that led to this work. As a result, we believe that AircraftVerse is sufficiently complex and interesting for both sets of researchers, who would see value in the data, while ensuring that they take into account all the limitations."
> > >
> > > > (3) As for the related work, the authors have not highlighted the significant value of the dataset.
> > >
> > > We have tried to highlight why our dataset is unique and how it varies significantly from previous datasets. In addition to our related work section, we also directly show a comparison in Figure 1, where we show that "Existing CAD datasets (SketchGraphs [27], DeepCAD [34], ABC [18]) are focused on CAD for mechanical parts" whereas AircraftVerse is a CPS dataset which offers much more complexity than any previous datasets, through its inclusion of multiple modalities and representations of a design.
> > >
> > > Thanks once again, and please consider these comments and additional material during the reviewer discussion.

---

### Official Review · Reviewer_prSM · 2023-08-02
**A Multimodal Dataset of Aerial Vehicle Designs**

**Rating:** 7
**Confidence:** 2
**Correctness:** Yes

**Strengths:**

- The creation of the AircraftVerse dataset is a significant contribution. The dataset contains 27,714 diverse air vehicle designs, making it the largest corpus of engineering designs with such complexity. This provides a valuable resource for researchers and engineers working in the field of aerial vehicle design.

- The authors have made efforts to capture the multimodal nature of aircraft design. Each design in the dataset includes a symbolic design tree, a Standard for the Exchange of Product (STEP) model data, a 3D CAD design, a 3D point cloud representing the design's shape, and evaluation results from high-fidelity state-of-the-art physics models. This level of detail allows for a more nuanced analysis and understanding of the designs.

- The paper presents baseline surrogate models for predicting design performance. This not only demonstrates the potential usage of the dataset, but also provides a starting point for researchers who want to utilize this dataset for their own studies.

- The dataset and the analysis presented in the paper span across several disciplines, including engineering, computer science, and machine learning. This underscores the potential of such work to drive interdisciplinary research and foster innovation in these fields.

**Additional Feedback:**

N/A

**Clarity:**

The paper is well-writen.


**Documentation:**

There is a GitHub README file. More detailed documentation is appreciated.

**Limitations:**

The authors did not explicitly discuss the limitations of this work in their paper. See the "Opportunities For Improvement" for my suggestions.

**Opportunities For Improvement:**

- Better documentation: Although the paper introduces the AircraftVerse dataset, it does not provide very detailed documentation to help the users get started. Some concrete examples are helpful.

- Lack of external validation: The paper does not mention any validation of the results or the surrogate models with external datasets or independent experts. Without external validation, it's difficult to assess the generalizability of the findings.

- Other baseline models: The paper mentions that the GCNN model is better at inferring structural interferences between components as it directly incorporates 3D information. However, PointNet-based models [1] are more popular in the point cloud-based deep learning community. It would make more sense to add a PointNet-based baseline.

[1] Qi, Charles R., et al. "Pointnet: Deep learning on point sets for 3d classification and segmentation." Proceedings of the IEEE conference on computer vision and pattern recognition. 2017.

**Relation To Prior Work:**

Yes

**Summary And Contributions:**

This paper presents AircraftVerse, a novel publicly available dataset of aerial vehicle designs. This dataset includes various modalities of representation, reflecting the multidisciplinary nature of aircraft design. It is the largest corpus of engineering designs with this level of complexity, containing 27,714 diverse air vehicle designs.

The authors also present baseline surrogate models that use different modalities of design representation to predict design performance. They conducted experiments to predict performance statistics from the design sequences and found that the sequential design description is useful for certain metrics, such as a design's ability to fly. However, the baseline Graph Convolutional Neural Network (GCNN) model performed better in inferring structural interferences between components, as it directly incorporates 3D information.

The contributions of this paper include:

- The introduction of the AircraftVerse dataset, a large-scale multimodal dataset of aerial vehicle designs. This dataset fills a significant gap in publicly available datasets for complex cyber-physical systems.
- The demonstration of the use of baseline surrogate models to predict design performance. These models serve as a starting point for further research.
- The presentation of experimental results that illustrate the effectiveness of different modalities of design representation in predicting performance metrics.

---

> ### Author Response · Authors · 2023-08-14
> **Response to prSM**
>
> Thanks very much for the feedback. As a result we have worked on the following:
> > Better documentation
>
> Please see the README.md has since been updated on how to get started with the dataset. Please let us know if this can be improved.
> > Lack of external validation
>
> We note that the authors of the simulation pipeline include domain experts in cyber physical systems (CPS) design and Aircraft design. We refer to Walker et al. 2022 and Bapty et al. 2022 for validation of the pipeline. These authors played a key role in the validation of the dataset.
> > Other baseline models
>
> Thanks to your suggestion of PointNet, we have since implemented this baseline and included the results in the updated paper.

---

> > ### Comment · Reviewer_prSM · 2023-08-31
> >
> > Thank you for your response. I don't have further concerns.

---

### Official Review · Reviewer_HgCT · 2023-08-02
**An interesting dataset, although the number of considered methods is limited**

**Rating:** 7
**Confidence:** 4
**Clarity:** the paper is well written

**Strengths:**

- Interesting, non-standard dataset: multi-modal data (3D data, symbolic representations data, etc.)
- Specially selected physically feasible designs
- Pre-calculated physical characteristics of the design

**Additional Feedback:**

All comments about improvements are provided above.

**Correctness:**

- the claims made are correct
- the dataset is interesting and there can be different types of problems based on this dataset
- the benchmark is limited

**Documentation:**

yes

**Ethics:**

I do not suspect any ethical concerns.

**Limitations:**

- The authors consider only a specific domain devoted to specific type of designs of  Electric Vertical Take-Off and Landing aircrafts
- Limited bechmark. For example, for point clouds there exists dozens of point-net like architectures. However, the authors considered only one example of such type of architectures
- It is not clear what is the exact purpose of the benchmark. Which class of algorithms do the authors want to benchmark?
- Are there any examples of solving design optimization problem using the benchmark?
Do the author plan to consider such type of applications to compare different methods?
- Are there any real designs of aircrafts in the database? If we start the design optimization process from such designs there are more chances to get something feasible as the resulting optimized design

**Opportunities For Improvement:**

- I would propose to specify more clearly what types of problems the authors would like to address using the existing benchmark
- The number of considered baselines in the benchmark is too small. I would propose to add more recent baselines
- Usually, a standard way to use such data set is to construct a surrogate model to make the design space exploration faster. So I would propose to consider some design space exploration problem

**Relation To Prior Work:**

It is clearly discussed how the work differs from previous approaches

**Summary And Contributions:**

- The authors developed an aerial vehicle design dataset, containing 27 714 diverse air vehicle designs. Each design consists of a symbolic design tree describing topology,  propulsion subsystem, battery subsystem, and other design details; a 3D CAD design; a 3D point cloud for the shape of the design; and target values from some physics models characterizing performance metrics such as hover time, etc.

- The authors also considered several basic surrogate models predicting various flight characteristics.

====

After reading the authors' comments and discussions with other reviewers I decided to increase the rating.

---

> ### Author Response · Authors · 2023-08-14
> **Response to reviewer HgCT**
>
> Thanks for the feedback. As a result we have made the following changes:
>
> > The number of considered baselines in the benchmark is too small.
>
> We have since added the additional baseline of PointNet and have updated the paper on OpenReview. The results are consistent with GCNN and further strengthen the conclusion that the point cloud modality is better for capturing 3D information. Thanks for the suggestion!
>
> > It is not clear what is the exact purpose of the benchmark.
>
> We have since elaborated on this in the additional limitations section by discussing the target audience. Furthermore, we would like to highlight that our paper is foremost a dataset paper. The baseline results are intended to be illustrative of the datasets potential, which we believe to be much broader than what we have been able to show.
>
> Furthermore, we refer to the summary response where we have incorporated the above feedback into the limitations section.

---

### Author Response · Authors · 2023-08-14
**Summary response to all reviewers**

Thanks very much for the constructive feedback on the paper. We have updated the paper on OpenReview and the github repo. We will respond individually to each review for specific points but will also refer back to this “summary response” to cover common suggestions as highlighted by the all reviewers:
* **Addition of Limitations Section** (prSM, vKs1, v2Uf, TaAN): We have used our additional page to write a limitations section. This section covers:
	* Structural limitations of the design language.
	* Limitations of the simulator.
	* Expands on how these limitations affect the target audience of the dataset, where we highlight the value of this dataset as useful for machine learning researchers looking to explore a novel CPS dataset as well as domain experts looking to explore the potential of data-driven approaches for design.
* **Addition of Ethics Statement Section** (29GD, xfyw, nS1d): We now include a full ethics statement that highlights the intended use of the dataset as a purely technical one, while strongly condemning any misuse of the dataset. We would also like to highlight the existing procedures that are in place at SRI (a non-profit research organization) to ensure the responsible release of any data. Please see the updated paper.
* **README.md has been updated** (prSM, vKs1): Thanks to feedback we have added a detailed section on how to use the dataset. Please see https://github.com/SRI-CSL/AircraftVerse. Please let us know if this can be improved.
* **Released Probabilistic Generator** (v2Uf, TaAN, 29GD): We agree that access to the probabilistic generator was important and have now included it in the github repo under `prob_gen`. This folder comes with its own README.md instructions to enable users to generate their own designs and explore the design language.
* **PointNet Baseline added** (prSM, HgCT): We have now added PointNet as an additional baseline at the suggestion of  prSM and HgCT. These results are given in Table 1 and Figure 6 of the updated paper on OpenReview.

We hope that the above improvements further strengthen what we believe to be generally favorable reviews.

---

> ### Author Response · Authors · 2023-08-24
> **Comment to Reviewers**
>
> Hi,
>
> Since the author-reviewer response period ends on August 29th, could you let us know if there are any outstanding comments regarding our response to the feedback? Please consider our responses if you would like to update your score or confidence ratings.
>
> Thanks so much for your time!

---

### Decision · Program_Chairs · 2023-09-22

**Decision:**

Accept (Poster)

**Comment:**

This paper presents a large dataset of aircraft assets (over 27K) and contains multi-modal representations for each model (e.g., STL, Point Cloud, STEP, etc.). The models have complicated structures and compositions of parts for making aircraft. The authors have done preliminary experiments using the proposed dataset on the task of predicting the performance characteristics of an aircraft given symbolic design trees as inputs. While there are some concerns regarding the discussion of limitations (which has been well addressed by the authors during rebuttal), broader applications of the dataset, and the fact that the models are probabilistically generated, most reviewers and AC agree on the contributions of this dataset since it fills in the gap of an important and interesting yet underexplored data domain, and its large scale and multi-modal richness, as well as demonstrating some examples for using the dataset. Therefore, an acceptance decision is recommended.